# Context and Diversity Matter: The Emergence of In-Context Learning in World Models

*†Fan Wang[1,2], *Zhiyuan Chen[1], *Yuxuan Zhong[1], Sunjian Zheng[1], Pengtao Shao[1], Bo Yu[1],
†Shaoshan Liu[1], Jianan Wang[4], Ning Ding[1], Yang Cao[2,3], and Yu Kang[2,3]

[1]Shenzhen Institute of Artificial Intelligence and Robotics for Society, Shenzhen, China
[2]University of Science and Technology of China, Heifei, China
[3]Anhui Province Key Laboratory of Intelligent Low-Carbon Information Technology and Equipment
[4]Astribot, Shenzhen, China

## Abstract

The capability of predicting environmental dynamics underpins both biological neural systems and general embodied AI in adapting to their surroundings. Yet prevailing approaches rest on static world models that falter when confronted with novel or rare configurations. We investigate in-context learning (ICL) of world models, shifting attention from zero-shot performance to the growth and asymptotic limits of the world model. Our contributions are three-fold: (1) we formalize ICL of a world model and identify two core mechanisms: environment recognition (ER) and environment learning (EL); (2) we derive error upper-bounds for both mechanisms that expose how the mechanisms emerge; and (3) we empirically confirm that distinct ICL mechanisms exist in the world model, and we further investigate how data distribution and model architecture affect ICL in a manner consistent with theory. These findings demonstrate the potential of self-adapting world models and highlight the key factors behind the emergence of EL/ER, most notably the necessity of long context and diverse environments. The codes are available at `https://github.com/airs-cuhk/airsoul/tree/main/projects/MazeWorld`.

## 1 Introduction

The ability to predict future environmental states is crucial for reasoning and decision-making in both animals and humans. Inspired by this principle, constructing predictive models, especially the world model (Ha & Schmidhuber, 2018b), to forecast environmental dynamics and outcomes forms the foundation for enabling agents to plan effective decisions and behaviors (Hafner et al., 2025; Zhang et al., 2023a; Mazzaglia et al., 2024; Samsami et al., 2024; Alonso et al., 2024). Consequently, world models are widely applied in fields such as navigation (Bar et al., 2025; Mendonca et al., 2021; Koh et al., 2021; Duan et al., 2024; Liu et al., 2025), autonomous driving (Hu et al., 2023; Russell et al., 2025; Gao et al., 2024; Zhang et al., 2024a; Wang et al., 2024c), robotics (Zhang et al., 2023b; Wu et al., 2023; Hansen et al., 2024; Pang et al., 2025; Zhou et al., 2024; Barcellona et al., 2025), and are considered a cornerstone of embodied artificial intelligence.

Despite their proven effectiveness across various applications, previous prediction frameworks largely rely on static world models optimized for zero-shot, few-shot, or instantaneous performance. In contrast, humans and animals achieve real-time adaptation (Vorhees & Williams, 2014) through predictive coding—a process where prediction errors drive attention, generate feedback, and motivate learning and adjustment (Rao & Ballard, 1999; Salvatori et al., 2023). For instance, when confronted with rare environments, humans experience surprise yet rapidly recalibrate their predictions for that setting, whereas static models continue to fail unless explicitly retrained on the relevant data. The ability to dynamically modify predictive mechanisms based on observational evidence, rather

---

*Equal Contribution
†Corresponding to: fanwang.px@gmail.com, shaoshanliu@cuhk.edu.cn

than relying solely on fixed parametric memory and external mapping modules, can effectively enable the model to adapt to environments unseen during training. This capability can be effectively addressed by In-Context Learning (ICL), as evidenced by recent advances in Large Language Models (LLMs) (Brown et al., 2020). However, existing investigations into ICL primarily focus on language-related tasks or simpler few-shot classification and regression tasks. ICL within world models, despite its potential importance, remains largely underexplored. Addressing this gap could complete the missing pieces of embodied AI, further improving generalization and self-adaptivity.

Following the Bayesian hypothesis of ICL (Panwar et al., 2023; Xie et al., 2024), we clarify two potential underlying mechanisms for ICL in world models: environment recognition (ER), which relies on parametric memory of the training environment, and environment learning (EL), which does not. By deriving upper error bounds for both ICL modes, we theoretically demonstrate that the emergence of ICL depends on environment diversity, complexity, and context length. This insight motivates the development of long-context adaptive world modeling. Consequently, we introduce the linear-attention long-context **world** model, *L2World*, which enables self-adaptation to environments through efficient memory updates within the context. Across cart-pole control and vision-based indoor navigation tasks, we empirically demonstrate that distributional properties and long-context capacity govern the ICL ability of world models. Despite employing lightweight image encoders and decoders in L2World, it establishes a new state-of-the-art for long-sequence observation prediction in cross-environment adaptation, outperforming methods that rely on computationally intensive diffusion-based image backbones. These results underscore the importance and potential of enhancing ICL through intentionally diversified datasets and long-context modeling architectures within world models, rather than focusing solely on immediate or zero-shot frame-level performance.

## 2 RELATED WORK

### 2.1 DYNAMIC PREDICTION AND WORLD MODELS

Dynamic models, also referred to as world models (Ha & Schmidhuber, 2018b; Forrester, 1958), encompass probabilistic, physical, or generative frameworks that formalize an AI system's environmental understanding (Sutton, 1990; Battaglia et al., 2013; Ha & Schmidhuber, 2018a; LeCun, 2022). These models predict future states by leveraging historical observations and play a pivotal role in advancing reinforcement learning (RL) methodologies and related domains. Specifically, they constitute foundational components in model-based RL (Finn & Levine, 2017; Schrittwieser et al., 2020), enable simulations that facilitate agent learning through virtual experiences (thereby reducing reliance on direct environmental interaction) (Hafner et al., 2019a; 2025; Samsami et al., 2024; Sutton, 1991; Kaiser et al., 2020), and serve as auxiliary tasks to augment model supervision (Jaderberg et al., 2016; Hu et al., 2022; Zhang et al., 2024c).

World models demonstrate robustness in integrating multi-modal raw sensor data, including visual, textual, inertial, and tactile inputs. To mitigate challenges posed by high-dimensional sensory inputs, representation learning paradigms such as Generative Adversarial Networks (GANs) (Goodfellow et al., 2014) and Variational Autoencoders (VAEs) (Kingma & Welling, 2013) are widely utilized to compress raw data into compact latent spaces. Subsequent temporal modeling in these reduced dimensions is achieved through latent world model architectures (Hafner et al., 2025; Wang et al., 2024d; Mazzaglia et al., 2024; Samsami et al., 2024; Hafner et al., 2019b; Zhang et al., 2024c; Garrido et al., 2024; Li et al., 2024), which capture sequential dependencies and enable coherent long-term predictions.

In navigation, early systems relied on traditional, hand-crafted pipelines such as SLAM. Recent work replaces these modules with generative world models, including diffusion (Bar et al., 2025), VAE (Koh et al., 2021), and RL-enhanced variants (Poudel et al., 2023; Duan et al., 2024), which reconstruct dynamics or simulate semantics (Nie et al., 2025; Liu et al., 2023). However, most existing methods disregard continual adaptation, especially across episodes, leaving a persistent gap between zero-shot performance and lifelong operation.

### 2.2 IN-CONTEXT LEARNING AND META-LEARNING

The approach based on parametric memory, which predominantly relies on gradient-based optimization or In-Weight Learning (IWL), has faced criticism for its lack of plasticity and the associated

challenges it poses in continual learning scenarios (Dohare et al., 2024). Conversely, ICL has emerged as a pivotal capability in large language models (Brown et al., 2020), facilitating generalization to novel tasks without the necessity of parameter fine-tuning. ICL leverages contextual memory for task solutions, rather than depending on parametric memory. The concept of ICL is not novel. Meta-learning (Santoro et al., 2016; Duan et al., 2016), which focuses on acquiring learning capabilities rather than mastering specific skills, utilizes well-curated environments or data instead of relying on large-scale, uncurated pre-training data. Nevertheless, the lack of well-structured, task-rich, and cost-efficient datasets continues to present a significant challenge.

ICL has been utilized to encode a diverse array of learning mechanisms, including language learning (Akyürek et al., 2024), regression (Garg et al., 2022), reinforcement learning (Laskin et al., 2023; Lee et al., 2023; Wang et al., 2025), and world models (Anand et al., 2022; Gupta et al., 2024), highlighting its resemblance to biological plasticity (Lior et al., 2024). Yet existing work concentrates on few-shot in-context adaptation, overlooking ICL's potential as contexts grow indefinitely. Concurrent studies have identified various mechanisms and circuits underlying ICL, including distinctions between task learning and task recognition (Pan et al., 2023), as well as retrieval versus inference (Park et al., 2025), among others. Key factors influencing the emergence of ICL have also been investigated, such as transience, task diversity, and context length (Chan et al., 2022; Anand et al., 2022; Wurgaft et al., 2025; Nguyen & Reddy, 2025), along with the relationship between IWL and ICL (Chan et al., 2025; Singh et al., 2025). However, these studies primarily focus on simplified regression and classification problems, while theoretical frameworks addressing the incentivization of ICL within world models remain underexplored.

## 3 METHODOLOGIES

We consider an environment $e$ specified by a Partially Observable Markov Decision Process (POMDP) $e : \langle O, S, A, T_e, Z_e \rangle$, where $S$ is the state space, $A$ is the action space, $O$ is the observation space, $T_e(s, a, s') = p_{\tau,e}(s'|s, a)$ is the transition model, and $Z_e(s, o) = p_{z,e}(o|s)$ is the observation model. [1] A fully observable MDP is denoted with $e : \langle S, A, T_e \rangle$ with $o \equiv s$. We denote a world model with the following equation:

**World Model:** $\hat{o}_{t+1} \sim \hat{p}_\theta(\cdot|q_t) = f_\theta(q_t),$ with $q_t = (s_t, a_t)$ (MDP)

$$\text{or } q_t = (o_{t-\Delta t}, a_{t-\Delta t}, \ldots, o_t, a_t) \text{ (POMDP) .} \quad (1)$$

Let $\theta$ denote the model parameters; values marked with a hat denote predictions, and unmarked values denote the ground truth. Consider extra contexts of observations and actions, $C_T = (o_1^{(C)}, a_1^{(C)}, \ldots, o_T^{(C)}, a_T^{(C)})$, where $T$ indexes the context length; the ICL capability of the world model is then characterized by the following condition:

$$\forall T_1 > T_2, D[\hat{p}_\theta(\cdot|q_t, C_{T_1})||p_e(\cdot|q_t)] < D[\hat{p}_\theta(\cdot|q_t, C_{T_2})||p_e(\cdot|q_t)], \quad (2)$$

Here, $D$ represents a metric measuring the error between two distributions (lower values indicate better performance). Notably, the ICL of the world model fundamentally relies on cross-episode contexts rather than intra-episode state estimation. To distinguish these concepts clearly, we use $q_t$ to denote short-term state estimation and $C_T$ to represent long-term ICL. While these are typically aligned in a single sequence in practice, maintaining this distinction facilitates rigorous theoretical analysis. Building on prior work that partitions in-context learning into different modes (Kirsch et al., 2022; Pan et al., 2023; Park et al., 2025), theoretically, we are able to identify two analogous modes within world-model ICL: *Environment Recognition* (ER) and *Environment Learning* (EL). We then derive error bounds that characterize the conditions under which each mode emerges.

### 3.1 ENVIRONMENT RECOGNITION (ER)

To clarify Equation (2), we consider a world model optimized on the finite environment set $\mathcal{E} = \{e_1, \ldots, e_{|\mathcal{E}|}\}$. Assume the system possesses an environment-specific model $\hat{p}_{\theta,e}$ for every

---

[1] While most prior work integrates the reward model into the world model, our analyses omit explicit consideration of rewards. Nonetheless, since rewards can typically be derived from the state or observation, our framework allows for straightforward extension to incorporate reward-related considerations.

environment $e$, then $\hat{p}_\theta$ decomposes as follows:

$$\hat{p}_{\theta,ER}(o_{t+1}|q_t, C_T) = \sum_{e \in \mathcal{E}} \underbrace{\hat{p}_\theta(e|q_t, C_T)}_{\substack{\text{Environment} \\ \text{Recognition}}} \cdot \underbrace{\hat{p}_{\theta,e}(o_{t+1}|q_t)}_{\substack{\text{Environment-Specific} \\ \text{World Model}}} \tag{3}$$

Therefore the ICL in world models arises mainly from the recognition of seen environments, while environment-specific world model are static within the inference. The environment-specific world model can be further decomposed into:

$$\hat{p}_{\theta,e}(o_{t+1}|q_t) = \int \underbrace{\hat{p}_{\theta,s,e}(s_t|q_t)}_{\text{State Estimation}} \cdot \underbrace{\hat{p}_{\theta,\tau,e}(s_{t+1}|s_t, a_t)}_{\text{Dynamics}} \cdot \underbrace{\hat{p}_{\theta,z,e}(o_{t+1}|s_{t+1})}_{\text{Observation Model}} ds_t, \tag{4}$$

where the estimate $\hat{p}_{\theta,e}$ is obtained by marginalizing over the terms that include state estimation, dynamics, and observation $(\hat{p}_{\theta,s,e}, \hat{p}_{\theta,\tau,e}, \hat{p}_{\theta,z,e})$. Therefore, in the ER regime, the model first acquires world models for the entire environment set through IWL or parametric memory, and then uses the context solely to identify the current environment.

## 3.2 Environment Learning (EL)

Equation (3) is efficient when the environment set $\mathcal{E}$ is small, yet its effectiveness diminishes rapidly as the size and diversity of $\mathcal{E}$ grow or when the system faces open worlds. However $\hat{p}_\theta$ can also be approximated without estimating $e$ at all, by directly accumulating the evidence for $(q_t, o_{t+1})$ across all contexts:

$$\hat{p}_{\theta,EL}(o_{t+1}|q_t, C_T) = \frac{p(q_t, o_{t+1}|C_T)}{p(q_t|C_T)} \tag{5}$$

An intuitive observation of Equation (5) is that EL functions, at minimum, as an in-context memorizer. For highly complex environments, its performance might degrade sharply, because accurate estimation of the transition of $q_t$ demands contexts that scale with the environment's complexity.

## 3.3 Theoretical Analyses of ER and EL

Although the training process and data distribution play a key role in effectively incentivizing ICL (Chan et al., 2022), how does the data distribution determine whether EL or ER emerges? If training consistently minimizes predictive error, the error bounds of EL and ER become the decisive factor in selecting the emergent mode. To investigate the conditions governing the emergence of the two modes (ER and EL), we analyze the error upper bounds for each paradigm. For tractable analysis, we introduce the following simplifying assumptions: (1) The observation, state, and action spaces are discrete; (2) Both modes achieve ideal state estimation $p_e(s|q)$; (3) The context $C_T$ has a uniform state-action distribution. Under these assumptions, we derive an upper bound on the error of the world model optimized over environment set $\mathcal{E}$, measured by the total variation (TV) distance, when deployed in an unseen environment $e_0$ at context horizon $T$. Formally, the TV error is bounded by:

**Theorem 1.** *For Environment Recognition and Environment Learning whose predictive models $\hat{p}_{ER/EL}$ have been sufficiently optimized on the training environments $\mathcal{E}$, the upper bound of the total-variation (TV) distance between the predicted and the ground-truth transition, given a context $C_T$ of length $T$, can be estimated as:*

$$TV(\hat{p}_{ER}, p_{e_0}) \leq \min[\underbrace{\alpha/3 \cdot (|\mathcal{E}| - 1) \cdot T^{-1/2}}_{\text{Recognition Error}}, \underbrace{\max_{e_1, e_2 \in \mathcal{E}} TV(p_{e_1}, p_{e_2})]}_{\text{Diversity}} + \underbrace{\min_{e \in \mathcal{E}} TV(\hat{p}_{\theta,e}, p_{e_0})}_{\text{Best Matching Error}}$$

$$TV(\hat{p}_{EL}, p_{e_0}) \leq \underbrace{\sqrt{2|O||S||A|log(4|O|/\delta)}}_{\text{Environment Complexity}} \cdot T^{-1/2},$$

$$\text{with probability } 1 - \delta, \text{ and } T > 4|S|^2|A|^2 \log(4|S||A|/\delta) \tag{6}$$

Proofs and detailed assumptions for the above theorems are deferred to Appendix A. An immediate observation from Theorem 1 is that EL enjoys an ideal error upper bound that decays as $T^{-1/2}$,

whereas ER carries a non-decaying residual term (the best-matching error) that becomes the dominant obstacle to generalizing across unseen environments. To enhance generalization, we therefore ask: under what condition is EL preferred to ER? For the entire training set $\mathcal{E}$, EL dominates whenever $\mathbb{E}_{e \in \mathcal{E}}\big[\mathrm{TV}(\hat{p}_{\mathrm{EL}}, p_e)\big] \ll \mathbb{E}_{e \in \mathcal{E}}\big[\mathrm{TV}(\hat{p}_{\mathrm{ER}}, p_e)\big]$; The opposite inequality favors ER. Although the errors themselves are intractable to evaluate directly, the following insights are obtained by comparing their upper bounds:

(1) **Lower environmental complexity and a greater number of environments favor EL over ER**: Note that the best-matching error is effectively zero because the model is evaluated only on environments seen during training. The cardinality of the training set, $|\mathcal{E}|$, affects only the ER bound, whereas the environmental complexity, $|O||S||A|$, influences only the EL bound. Consequently, lower complexity combined with a larger training set pushes the EL bound below the ER bound.

(2) **Long context and environment diversity are key to both ER and EL**: As the upper error bound of ER effectively approaches zero when diversity is low, the emergence of both ER (where the identification error would never dominate) and EL is precluded. Once the training set is sufficiently diverse, both ER and EL obtain an upper error bound that decays as $T^{-1/2}$, demonstrating that long context is indispensable for either mechanism.

(3) **Over-training and powerful IWL facilitate ER over EL**: we hypothesize that IWL perfectly models the environment-specific dynamics ($\hat{p}_{\theta,e}$) in the training set, so the best-matching error is nearly zero during training; however, this is not always true. Early in training, IWL can still incur large errors, transiently pushing the model toward EL instead of ER. This transiency is also investigated by prior works in ICL (Singh et al., 2023; 2025). As training proceeds and IWL becomes increasingly accurate, the model may revert to ER.

In the following section, we empirically confirm that these insights hold not only for discrete settings but also for continuous MDPs and POMDPs. Because theory predicts that large environmental diversity and low task complexity are required for incentivizing EL, and most of the closed benchmarks can not satisfy those requirements. We construct our dataset from randomly sampled *cart-poles* and procedurally generated *mazes* to evaluate the performance of ER and EL.

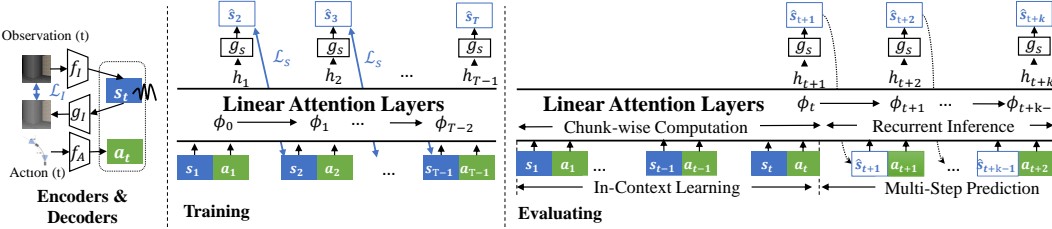

Figure 1: The world model structure for the empirical study.

### 3.4 L2World: Long-Context and Linear-Attention World Models

Prior work relies on multi-token representations or diffusion models to deliver high-fidelity single-frame reconstructions specifically for images. While these approaches set the state-of-the-art for static images, they introduce prohibitive memory and computational bottlenecks when sequences grow to the length required for EL. We therefore introduce L2World that trades per-frame fidelity for temporal scalability: for image observations, we compress each frame $o_t$ into a latent state $s_t$ with a lightweight variational auto-encoder (VAE) (Kingma & Welling, 2014) whose encoder $f_I$ and decoder $g_I$ are ResNet stacks; for low-dimensional observations, we simply apply a small multi-layer perceptron encoder/decoder pair. For computational efficiency, we do not model the full state estimation with $\hat{p}(s_t|q_t)$. Instead, we construct a pseudo-state that depends solely on the instant observation and leaves all temporal-related encoding to transition modeling (Mazzaglia et al., 2024). We then construct the adaptive world model $\hat{p}_\theta(o_{t+1}|q_t, C_T)$ using an efficient sequence decoder $f_\theta$. Here we implement gated slot attention layers (Wang et al., 2020; Yang et al., 2024; Zhang et al., 2024b) with chunk-wise parallelization during the training phase, while retaining the recurrent form during the inference phase. The predictor of transition at first yields the output $h_t$, which is further processed by the decoder $g_S$ to produce the predicted state $\hat{s}$, corresponding to predicting a Gaussian distribution over the latent space $\hat{p} \sim \mathcal{N}(\hat{s}_t, \sigma_s^2)$. Although this assumption could lead to significant loss of accuracy in stochastic environments, for the navigation tasks we consider, it is acceptable and

greatly increases computational efficiency. The model and the target function are listed as follows (see details in Appendix B.2):

$$\text{Observation Encoder}: s_t, \sigma_{s,t} = f_I(o_t) \quad \text{Observation Decoder}: \hat{o}_t = g_I(\hat{s}_t)$$

$$\text{Latent Decoder}: \hat{s}_t, \hat{\sigma}_{s,t} = g_S(h_t) \quad \text{Action Encoder}: a_t = f_A(Action[t])$$

$$\text{Chunk-wise Temporal Modeling}: \phi_t, h_1, ..., h_t = f_\theta(s_1, a_1, ..., s_t, a_t)$$

$$\text{Recurrent Temporal Modeling (Evaluating)}: \phi_t, h_{t+1} = f_\theta(\phi_{t-1}, s_t, a_t)$$

$$\text{Observation Reconstruction Loss}: \mathcal{L}_o = ||o_t - g_I(\hat{s}_t)|| + \lambda KL(\mathcal{N}(s_t, \sigma_{s,t})||\mathcal{N}(0,1))$$

$$\text{State Transition Loss}: \mathcal{L}_s = -\sum_t KL(\mathcal{N}(s_t, \sigma_{s,t})||\mathcal{N}(\hat{s}_t, \hat{\sigma}_{s,t}))$$

When the observation is an image, we first pre-train the image encoder $f_O$ and decoder $g_O$ on pre-sampled observations; after this stage, their parameters are frozen while the temporal model is trained. For lower-dimensional observations, all encoders/decoders are updated jointly with the temporal model in a single end-to-end phase.

## 4 EXPERIMENTS

Although prior studies (Chan et al., 2022; Singh et al., 2023; Raventós et al., 2023) have validated the influence of data distribution on ICL, they have largely concentrated on simplified tasks such as regression and classification. To examine how data distribution, model architecture, and training procedure jointly affect the emergence of EL/ER, we select two canonical benchmarks. First, the cart-pole, a classical continuous-control problem, in which EL primarily targets the acquisition of varying embodiments and physical constants. Second, indoor navigation, a widely recognized POMDP, in which EL focuses on learning and memorizing spatial coefficients. These two experiments confirm not only the impact of each factor on EL but also demonstrate that EL spans a broad generalization spectrum, extending from spatial reasoning and memorization to adaptation of embodiment and physical constants.

### 4.1 RANDOM CART-POLES

**Experiment Setting**. To investigate EL and ER in cart-pole environments, we randomized four variables in the environment settings: gravity $g$, cart mass $m_c$, pole mass $m_p$, and pole length $l$. We focus on two different scopes of the configurations to investigate the impact of the diversity issue: Scope 1 remains close to the original task, whereas Scope 2 covers a larger region and excludes Scope 1 (details are left to Figure 6). For each environment, we first trained an RL agent and then collected trajectories with an expert policy perturbed by uniform noise spanning $[0.3, 0.7]$ to ensure adequate coverage of the state-action space. We trained five comparison models; all share the same data scale (128K trajectories $\times$ 200 step/trajectory) but differ in the number and scope of the environments.

- 1-Env: 96K trajectories are sampled from the original environment ($g = 9.8, m_c = 1.0, m_p = 0.1, l = 0.5$), with 200 steps per trajectory.
- 4-Envs: 4 environments sampled from Scope 1+2, each with 24K trajectories.
- 16-Envs: 16 environments sampled from Scope 1+2, each with 8K trajectories.
- 8K-Envs (Scope 1/1+2): 8,000 environments sampled from Scope 1 or 1+2, each with 16 trajectories.

We evaluate with three test sets: (1) Seen 4-Envs, but the trajectories are independently sampled; (2) 256 environments from Scope 1; and (3) 256 environments from Scope 2. We evaluate mainly with the average prediction error, which is averaged over each of the context lengths $T$.

We plot the evaluation results in Figure 2, where the following insight is worth noting:

**Importance of both environment scope and environment number**: A comparison between the model trained on 1 environment (1 Env) and 4 environments (4 Envs) with the other group demonstrates that an insufficient number of environments leads to the absence of ICL and generalization, except in the tasks that the model has already seen. The 4 Envs group exhibits clear ER characteristics,

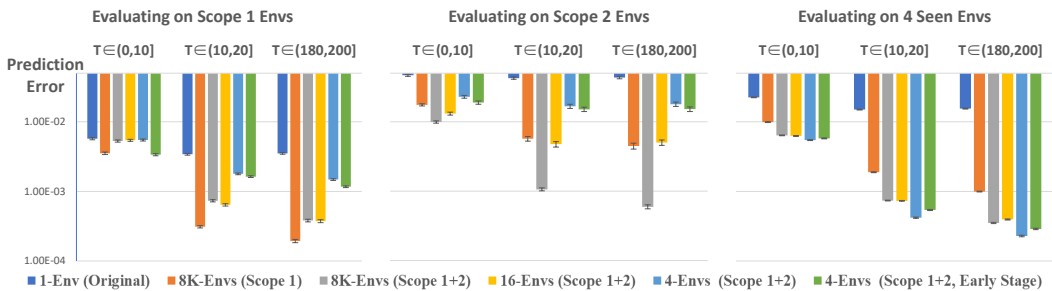

Figure 2: Comparison of models trained on different datasets (color-coded) in Cart-Poles. Performance varies markedly with the training data, revealing distinct tendencies toward ER, EL, or an inability to perform ICL.

with a substantial performance gap between seen and unseen tasks. The 16 Envs (Scope 1 + 2) and 8K Envs (Scope 1) groups display similar capabilities; however, they lag significantly behind the 8K Envs (Scope 1 + 2) group, indicating that both the scope of tasks and the number of tasks are crucial.

**Divergence between few-shot and many-shot performances**: Another insight gleaned from the comparison between 4-Envs and 8K-Envs is that the latter, which has a broader generalization scope, also requires more context to learn. This is evidenced by the fact that the performance of the latter group does not surpass that of the former group until $T > 10$. This also validates the theoretical analysis, indicating that a longer context is a cost for achieving better generalization.

**Over-training reduces generalization when training environments are insufficient**: To isolate the effect of over-training, we extract an early-stage checkpoint from the 4-Env group and evaluate it across all environments. Although its performance in seen environments is sub-optimal, it generalizes to unseen environments with a considerable margin over the over-trained model, confirming the shift from ICL- to IWL-based reliance.

**EL exhibits a smaller generalization gap and a relatively lower upper error bound.** According to Equation (6), as the context length increases, the Best Matching Error (BME) remains within the upper error bound in ER, thereby limiting generalization. In contrast, EL is not affected by this term. To further investigate these differences on a case-by-case basis, we examine the correlation between the BME of each of the 130 unseen test cart-poles and their corresponding prediction errors. The BME for each model and testing environment is estimated by applying the ground-truth world model of each environment in the training set to the test environment and selecting the minimum error. Note that we plot the prediction error for $T > 100$; therefore, the terms containing $T^{-1/2}$ are negli-

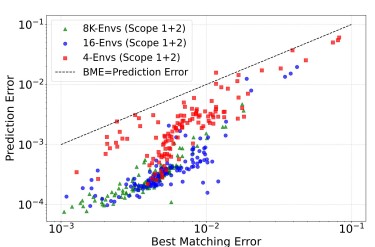

Figure 3: Best Matching Error (BME) versus prediction error for various models across 130 test cart-poles.

gible, reflecting asymptotic performance. The results, shown in Figure 3, reveal that the model trained with only four environments exhibits upper error bounds close to the line error = BME, whereas the upper error bound progressively moves below this line as the number of training environments increases. This not only validates the BME as a crucial term in Equation (6) but also confirms a transition from ER to EL mode as the number of environments increases.

## 4.2 NAVIGATION

**Experiment settings**. Procedurally generated mazes are a demanding test-bed for transition prediction (Pašukonis et al., 2023; Wang et al., 2024a). By stripping observations of semantic cues, the maze framework keeps task complexity low while still exposing models to stochastic, partially observable dynamics. We amplify diversity through fully randomized configurations that vary topology, textures,

Table 1: A summary of data distribution across the training datasets.

| Training DataSet | # envs ($|\mathcal{E}|$) | Len. Traj. | # Traj. | # frames | Indoor area |
|---|---|---|---|---|---|
| Maze-32K-L | 32K | 10K | 32K | 320M | $380 \sim 3422m^2$ |
| Maze-32K-S | 32K | 100 | 3.2M | 320M | $380 \sim 3422m^2$ |
| Maze-128-L | 128 | 10K | 32K | 320M | $380 \sim 3422m^2$ |
| Maze-128-S | 128 | 100 | 3.2M | 320M | $380 \sim 3422m^2$ |
| ProcTHOR-5K | 5K | 2K | 5K | 10M | $40 \sim 600m^2$ |
| ProcTHOR-40K | 40K | 2K | 40K | 80M | $40 \sim 600m^2$ |

object placement, and agent embodiment [1]. Room-tour data are collected in a way similar to cart-pole: we perturb the oracle object-navigation policy (Ehsani et al., 2024; Pašukonis et al., 2023) with random noise levels in $[0.05, 0.95]$ and record the complete trajectory of an agent in each environment. The oracle policy was derived using Dijkstra's algorithm based on the ground truth 2-D occupancy maps. Observations are RGB images standardized to $128 \times 128$ pixels; The action space comprises 17 discrete actions, each corresponding to a unique offset and rotational movement. Table 1 lists the resulting training sets, each drawn from a different number of environments and exhibiting varying trajectory lengths. To isolate the effect of data distribution, all maze datasets contain the same number of frames but differ in the coverage of trajectories and environments. To further investigate transferability to more realistic environments, we also collect trajectories from the semantically rich ProcTHOR simulation, which offers a wide variety of assets (Kolve et al., 2017; Deitke et al., 2022). Specifically, we curate two datasets: a larger one with 40,000 trajectories and a smaller one with 5,000 trajectories, each trajectory having a length of 2,000 frames.

For training, inspired by overshooting (Hafner et al., 2019b) and Hu et al. (2022), we randomly mask the $s_t$ of the input sequences at some positions to enhance the model's capability to predict the distant future (see training details in Appendix B.1). Evaluation is conducted on both seen and unseen tasks. By default, we use an evaluation set scale of $|\mathcal{E}| = 256$. The evaluation process involves encoding a context of length $t$ for EL and then predicting future $k$-step transitions using auto-regression and the ground truth action records $a_{t+1}, a_{t+2}, ..., a_{t+k}$. We assess the error in both the latent spaces and the decoded images, which we refer to as $k$-step prediction with context length $T = t$. For example, $T = 10$ and $k = 4$ predict future 4 steps with a context length of 10.

We evaluate two additional baselines alongside the proposed long-context world model: (1) Navigation World Model (NWM) (Bar et al., 2025), which employs diffusion layers to predict the next frame from the preceding four frames; (2) Dreamer-v3 (Hafner et al., 2019a), which uses LSTM layers for temporal encoding. For NWM, we retain its original pre-trained image encoders and re-train only the diffusion layers on the target dataset. For Dreamer-v3, we remove the policy components and train only the world-model module to ensure a fair comparison.

Table 2: Comparison of the performances (PSNR ↑) of 1-step future prediction in Mazes.

| Model | Seen | | | | | Unseen | | | | |
|---|---|---|---|---|---|---|---|---|---|---|
| | T=1 | T=10 | T=100 | T=1000 | T=10000 | T=1 | T=10 | T=100 | T=1000 | T=10000 |
| L2World (Maze-32K-L) | 16.80 | 20.97 | 23.11 | 24.65 | 25.05 | 16.37 | **21.24** | **23.17** | **24.66** | **24.65** |
| L2World (Maze-32K-S) | 18.57 | 19.28 | 19.67 | 20.21 | 20.48 | **18.45** | 19.24 | 19.63 | 20.29 | 20.31 |
| L2World (Maze-128-S) | 19.47 | 20.39 | 20.58 | 22.02 | 21.77 | 18.01 | 18.63 | 19.00 | 19.67 | 19.63 |
| L2World (Maze-128-L) | 18.54 | 20.86 | **23.32** | **25.65** | **26.00** | 17.54 | 19.43 | 20.96 | 21.54 | 21.52 |
| Dreamer (Maze-32K-L) | 16.40 | **21.82** | 19.24 | 21.26 | 21.89 | 16.81 | 20.48 | 21.40 | 22.65 | 22.12 |
| Dreamer (Maze-128-L) | 17.13 | 20.64 | 21.83 | 22.20 | 22.43 | 14.26 | 14.54 | 14.09 | 13.46 | 13.50 |
| NWM (Maze-32K-L) | **20.84** | 20.21 | 19.19 | 22.32 | 21.06 | 16.20 | 16.71 | 17.00 | 17.37 | 17.85 |

**Impact of data distribution and model architecture on ICL.** Table 2 reports the next-frame prediction quality estimated by PSNR ($k = 1$) for models trained on different datasets. Three empirical findings corroborate Theorem 1: (1) The 32K-L dataset yields the best generalization to unseen environments, whereas the 128-L dataset excels on seen ones; in both cases, peak performance occurs at the asymptotic stage, not at the beginning of the context. (2) Long-context training consistently produces stronger ICL than short-context training, confirming that extensive context is

---

[1] Procedural Maze environments:
https://github.com/FutureAGI/Xenoverse/tree/main/xenoverse/mazeworld

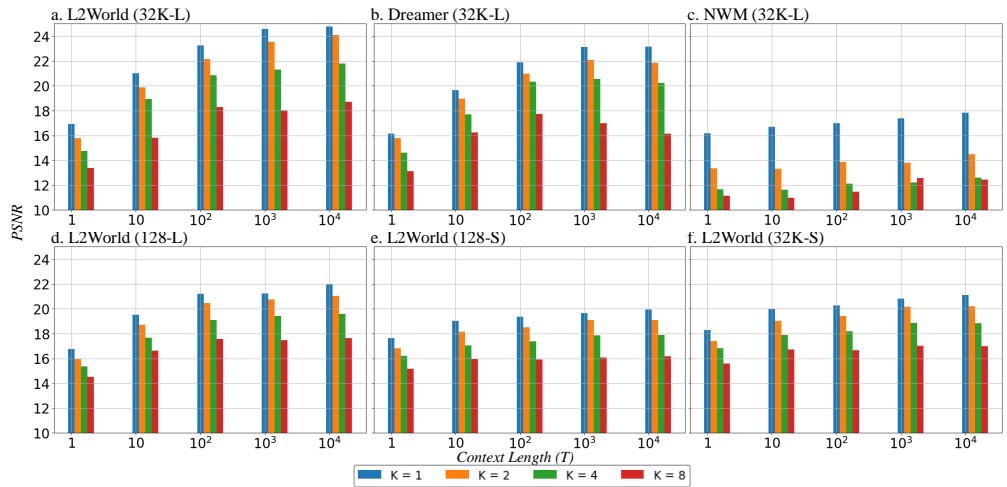

Figure 4: Comparison of k-step autoregressive PSNR in Mazes(Unseen).

necessary for ICL to emerge. (3) Dreamer and NWM fall short even with a long-context dataset: Dreamer's LSTM backbone and NWM's 4-frame horizon show that architectures incapable of fully leveraging long contexts cannot achieve many-shot ER. Figure 4 further presents the $k = \{1, 2, 4, 8\}$-step prediction performances on unseen Mazes, measuring how far ahead the world models can reliably foresee. For $k > 1$, the performance–context-length curve largely tracks the next-frame trend across models, except for Dreamer (Maze 32K-L): at $k = 8$, its performance plateaus once $T > 100$, whereas $k = 1$ keeps improving, revealing a larger compound-error accumulation than in our method.

Table 3: Comparison of the PSNR of 1-step future prediction in ProcTHOR (Unseen)

| Model | Pre-train | Post-train | T=1 | T=10 | T=100 | T=1000 | T=10000 |
|---|---|---|---|---|---|---|---|
| L2World | - | | 15.49 | 18.22 | 19.02 | 19.74 | 19.81 |
| L2World | Maze-32K-L | ProcTHOR-5K | 16.46 | **20.23** | **21.05** | **21.89** | **22.04** |
| L2World | Maze-32K-S | | **19.80** | 19.45 | 19.86 | 20.57 | 20.61 |
| L2World | Maze-128-L | | 19.16 | 19.60 | 20.20 | 20.94 | 16.46 |
| Dreamer | - | | 19.82 | 22.61 | 23.99 | 23.51 | 22.76 |
| NWM | - | ProcTHOR-40K | 18.30 | 21.41 | 21.11 | 21.02 | 20.08 |
| L2World | - | | **21.57** | 22.67 | 23.39 | 24.92 | 22.98 |
| L2World | Maze-32K-L | | 17.21 | **22.81** | **24.32** | **25.40** | **23.94** |

**EL transfers better than ER.** In Table 3, we train L2World on Proc-THOR trajectories and evaluate it on unseen ProcTHOR scenarios. The EL model pre-trained on Maze-32K-L not only excels in unseen mazes but also maintains its advantage when fine-tuned on the small ProcTHOR-5K dataset. Its transferability significantly surpasses that of Maze-128-L and other baselines, demonstrating EL's domain generality. Further increasing the amount of ProcTHOR data leads to continuous improvement

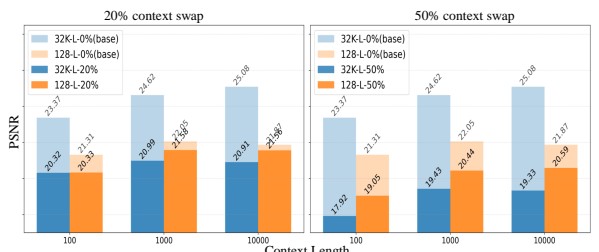

Figure 5: The decline in performance of EL (trained with Maze-32K-L) and ER (trained with Maze-128-L) when observations in contexts are shuffled, measured by PSNR.

in our model while preserving a substantial margin over Dreamer and NWM. However, performance at $T = 1K$ to $T = 10K$ begins to deteriorate when the ProcTHOR training data increases from 5K to 40K, suggesting that insufficient-length data ($T \leq 2K$) impairs the long-ICL ability acquired in Maze scenarios.

**EL is more sensitive to context perturbations than ER.** We investigate how models trained on datasets with varying levels of diversity (Maze-32K-L versus Maze-128-L) respond to perturbations

in context. Specifically, to assess the importance of context on performance, we randomly shuffle 20% or 50% of the observations within contexts while keeping the actions unchanged. The results are illustrated in Figure 5. Interestingly, we find that models trained on Maze-32K-L (which are expected to exhibit EL) are more severely affected by these perturbations than those trained on Maze-128-L. This suggests that EL depends more heavily on context, whereas ER relies more on model parameters and is therefore less influenced by changes in context.

## 5 CONCLUSIONS AND LIMITATIONS

**Conclusions**: This work investigates in-context learning of world models, specifically dynamic models, focusing on the possible modes of EL and ER in MDP and POMDP. Theoretically, we analyze error upper bounds for both modes to characterize their properties and identify the conditions under which each excels. Empirically, we introduce L2World and validate these insights in cart-pole and navigation tasks. Our results underscore that both high environment diversity and sufficient context length of the world model are essential to elicit EL.

**Limitations**: At present, our analysis is confined to the dynamic model; the reward and policy models can be addressed subsequently. This work constitutes a first step toward broader ICL mechanisms such as In-Context Reinforcement Learning. More sophisticated validations on real-world datasets and environments are desirable in the future.

### ACKNOWLEDGMENTS

This project is supported by Longgang District Shenzhen's "Ten Action Plan" for supporting innovation projects (Grant LGKCSDPT2024002), Guangdong Provincial Leading Talent Program (Grant No. 2024TX08Z319), National Natural Science Foundation of China (62033012), Guangdong S&T Program (Grant No.2025B0909040003), and the "Zhiguo" Action of Guangxi Science and Technology Program under Grant No.ZG2503980003.

### ETHICS STATEMENT

This research adheres to the ethical standards of the machine learning community. The datasets used in this work were synthetic and designed solely for academic research; they do not contain personal, sensitive, or identifiable information.

### REPRODUCIBILITY STATEMENT

We provide details of datasets and experiment settings in Section 4 and Appendix B. The training and evaluating codes are available at `https://github.com/airs-cuhk/airsoul/tree/main/projects/MazeWorld`. The benchmarking environments are available at `https://github.com/FutureAGI/Xenoverse/blob/main/xenoverse/metacontrol/random_cartpole.py` (random cart-pole) and `https://github.com/FutureAGI/Xenoverse/tree/main/xenoverse/mazeworld` (maze).

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

## A    Assumptions and Proofs for Theorem 1

We first define

$$\Delta(e_1, e_2) = \mathbb{E}_q D_{KL}(\hat{p}_{\theta,e_1}(\cdot|q), \hat{p}_{\theta,e_2}(\cdot|q))$$
$$\kappa(e_1, e_2) = \max_q D_{KL}(\hat{p}_{\theta,e_1}(\cdot|q), \hat{p}_{\theta,e_2}(\cdot|q))$$
$$\hat{e}_0 = argmin_{e\in\mathcal{E}}\Delta(e, e_0)$$
$$\kappa_i = \inf_{e\in\mathcal{E}, e\neq\hat{e}_0} \kappa(e, \hat{e}_0)$$
$$\kappa_s = \sup_{e\in\mathcal{E}, e\neq\hat{e}_0} \kappa(e, \hat{e}_0)$$
$$p_e(C_T) = \prod_{(q_t, o_{t+1})\in C_T} p_e(o_{t+1}|q_t)$$

We then make the following **Assumptions**:

- Queries $q_t = (s_t, a_t)$ are sampled i.i.d. from a distribution $\mu(s, a)$ with $\mu(s, a) \equiv \frac{1}{|S||A|}$.

- $\kappa(e_1, e_2) = \alpha(e_1, e_2)^2 \Delta(e_1, e_2)$ with $\alpha(e_1, e_2) = \sqrt{\frac{\kappa(e_1, e_2)}{\Delta(e_1, e_2)}}, \alpha > 1$. We further define $\alpha = \max_{e_1, e_2} \alpha(e_1, e_2)$, so that $\Delta(e_1, e_2) \geq \frac{\kappa(e_1, e_2)}{\alpha^2}$. Note that $\alpha$ can be interpreted as the measure of "non-uniformity" between any two environments in the set: it attains maximum when the two environments are almost identical yet differ significantly at only a few positions, and approaches 1 when the environments are either completely different or exactly the same.

- The environment recognizer selects closest task $\hat{e}$ by $\hat{e} = argmax_{e\in\mathcal{E}} p_e(C_T)$

**Proof for the first part of Theorem 1**: First, we gave that

$$TV(\hat{p}_{ER}, p_{e_0}) = TV(\hat{p}_{\theta,\hat{e}}, p_{e_0}) \leq TV(\hat{p}_{\theta,\hat{e}}, \hat{p}_{\theta,\hat{e}_0}) + TV(\hat{p}_{\theta,\hat{e}_0}, p_{e_0})$$
$$= TV(\hat{p}_{\theta,\hat{e}}, \hat{p}_{\theta,\hat{e}_0}) + \min_{e\in\mathcal{E}} TV(\hat{p}_{\theta,e}, p_{e_0}) \tag{7}$$

We then estimate the first term and use the Chernoff bound for derivation:

$$TV(\hat{p}_{\theta,\hat{e}}, \hat{p}_{\theta,\hat{e}_0}) = \sum_{e\in\mathcal{E}, e\neq\hat{e}_0} p(p_e(C_T) > p_{\hat{e}_0}(C_T)) TV(\hat{p}_{\theta,e}, \hat{p}_{\theta,\hat{e}_0})$$
$$\leq \sum_{e\in\mathcal{E}, e\neq\hat{e}_0} \alpha \cdot \underbrace{\exp(-T\cdot\Delta(e, \hat{e}_0))}_{\text{Chernoff bound}} \underbrace{\sqrt{1/2\Delta(e, \hat{e}_0)}}_{\text{Pinsker's Inequality}}$$
$$< \sum_{e\in\mathcal{E}, e\neq\hat{e}_0} \underbrace{\frac{\alpha}{2\sqrt{e\cdot T}}}_{\text{achieved maximum when } T\cdot\Delta(e,\hat{e}_0)=1/2}$$
$$< \frac{\alpha(|\mathcal{E}| - 1)}{3\sqrt{T}} \tag{8}$$

On the other hand, by definition, the TV of ER satisfies the following upper bound:

$$TV(\hat{p}_{ER}, p_{e_0}) \leq \max_{e_0\in\mathcal{E}} TV(\hat{p}_{\theta,e}, p_{e_0}),$$
$$\leq \max_{e_1, e_2\in\mathcal{E}} TV(p_{e_1}, p_{e_2}) + \min_{e\in\mathcal{E}} TV(\hat{p}_{\theta,e}, p_{e_0}). \tag{9}$$

By synthesizing Equation (7), Equation (8), and Equation (9), the proof is complete.

**Proof for the second part of Theorem 1**. We keep the aforementioned assumptions that the distribution of the context $C_T$ is uniform on the state and action space. We denote $n(s, a)$ as times of

appearance of $(s, a)$ in $C_T$. It is first straightforward to prove with Hoeffding's inequality that with probability of $1 - \delta/2$,

$$\frac{T}{|S||A|} - \sqrt{\frac{T \log (4|S||A|/\delta)}{2}} \leq n(s, a \in C_T) \leq \frac{T}{|S||A|} + \sqrt{\frac{T \log (4|S||A|/\delta)}{2}} \quad \forall s, a$$

Then, by add the constraint $T > 4|S|^2|A|^2 \log(4|S||A|/\delta)$, with at least probability of $1 - \delta/2$,

$$n(s, a) > \frac{T}{2|S||A|} \tag{10}$$

To estimate $\hat{p}_{EL}(s'|s, a)$, we use Equation (5) to acquire:

$$\begin{aligned}
\hat{p}_{\theta, EL}(o_{t+1}|q_t, C_T) &= \frac{p(q_t, o_{t+1}|C_T)}{p(q_t|C_T)} \\
&= \frac{\sum_s p(s, a_t, o_{t+1}|C_T)p(s|q_t)}{\sum_s p(s, a_t|C_T)p(s|q_t)}
\end{aligned} \tag{11}$$

Now employ Equation (10) and Equation (11) in the calculation of TV:

$$\begin{aligned}
TV(\hat{p}_{EL}, p_{e_0}) &= TV\left(\frac{\sum_s p(s, a_t, o_{t+1}|C_T)p(s|q_t)}{\sum_s p(s, a_t|C_T)p(s|q_t)}, \frac{\sum_s p(s, a_t, o_{t+1})p(s|q_t)}{\sum_s p(s, a_t)p(s|q_t)}\right) \\
&\leq \max_s TV\left(\frac{n(s, a_t, o_{t+1} \in C_T)}{n(s, a_t \in C_T)}, \frac{p(s, a_t, o_{t+1})}{p(s, a_t)}\right) \\
&\leq \underbrace{\sqrt{\frac{|O|log(4|O|/\delta)}{n(s, a_t \in C_T)}}}_{\text{Hoeffding's inequality}} \text{ with probability at least } 1 - \delta/2 \\
&< \sqrt{\frac{2|O||S||A|log(4|S|/\delta)}{T}} \text{ with probability at least } 1 - \delta
\end{aligned}$$

This finishes the proof of Theorem 1

## B  ADDITIONAL EXPERIMENT SETTINGS

### B.1  ENVIRONMENTS AND DATASETS

**Cart-pole**: Figure 6 lists the scopes of the Cart-Pole environment variants, their t-SNE visualization, and details of the training and evaluation data. All datasets share a trajectory length of 200, which is sufficient for ICL in Cart-Pole variants. The training data comprises a total of 25.6M timesteps to avoid interference from data scale on performance, and the evaluation data contains 205K steps in total.

**Mazes**: Our maze environment is closely related to the settings described in Pašukonis et al. (2023); Wang et al. (2024b), which are generated on a $15 \times 15$ grid world. The distribution of the sampled mazes is shown in Figure 7 and Figure 8. The primary distinction between our mazes and those in previous work is the enhanced environmental diversity achieved through randomized configurations, which include the following:

- The textures of the ceiling, ground, and walls are randomly selected from a collection of 87 real-world textures.
- The scale of each grid varies from $1.5\,\text{m}$ to $4.5\,\text{m}$, and the indoor height ranges from $2\,\text{m}$ to $6\,\text{m}$.
- The ground clearance and the field of view (FOV) of the camera are varied between $[1.6m, 2.0m]$ and $[0.3\pi, 0.8\pi]$, respectively.
- Two-wheeled dynamics are employed for the embodiment.
- Each environment involves $[5, 15]$ objects, each marked with a crossable, translucent light wall of a different color.

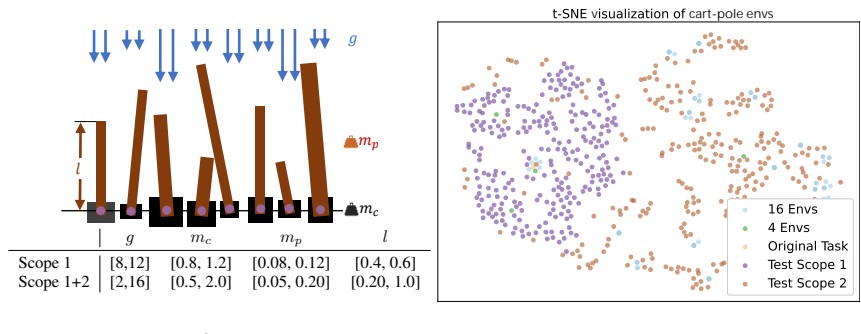

| DataSets | # envs ($|\mathcal{E}|$) | Len. Traj. | # Traj. | # time steps | Scope |
|---|---|---|---|---|---|
| 1-Env | 1 | 200 | 128K | 25.6M | Original |
| 4-Envs | 4 | 200 | 32K | 25.6M | Scope 1 & Scope 2 |
| 16-Envs | 16 | 200 | 8K | 25.6M | Scope 1 & Scope 2 |
| 8K-Envs (Scope 1) | 8K | 200 | 16 | 25.6M | Scope 1 |
| 8K-Envs (Scope 1+2) | 8K | 200 | 16 | 25.6M | Scope 1 & Scope 2 |
| Evaluation (Scope 1) | 256 | 200 | 4 | 205K | Scope 1 |
| Evaluation (Scope 2) | 256 | 200 | 4 | 205K | Scope 2 |
| Evaluation (4-Envs) | 4 | 200 | 256 | 205K | Scope 1 & Scope 2 |

Figure 6: Configuration scopes and cases of random Cart-Poles (upper left), t-SNE visualization of the configuration distribution (upper right), and a list of training and evaluation datasets.

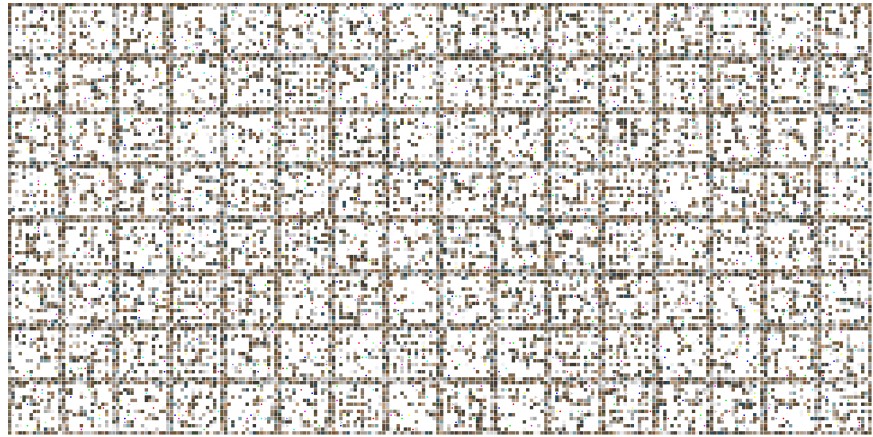

Figure 7: bird's-eye view of 128 procedurally generated mazes in the Homogeneous dataset.

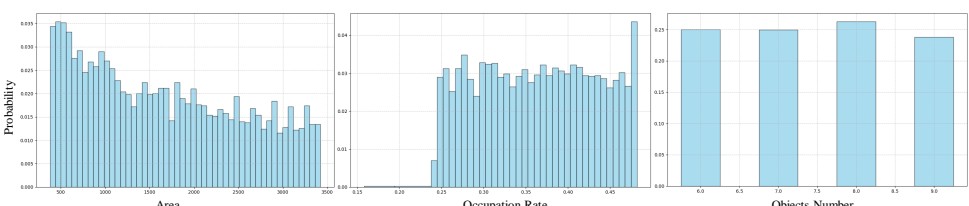

Figure 8: Distribution of the configurations (area, occupancy rate, and number of objects in each scene) in the $32K$ procedurally generated mazes

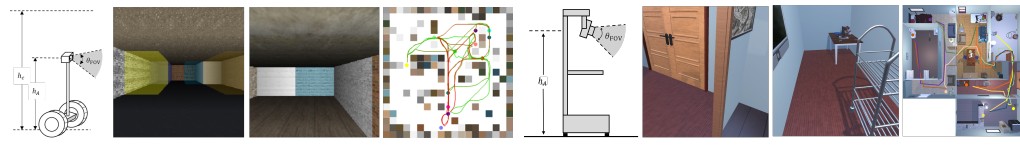

(a) Procedurally Generated Mazes                    (b) ProcTHOR

Figure 9: Illustration of the embodiment, the observation, and the trajectories in procedurally generated mazes (a) and ProcTHOR (b).

| Environments | Encoder & Decoder | $D$ | $L$ | $d$ | $l_M$ |
|---|---|---|---|---|---|
| Cart-pole | Raw | 128 | 4 | 4 | 64 |
| Navigation | Image | 1024 | 18 | 32 | 64 |

Table 4: Model architecture and hyper-parameters for the two classes of environments

- The agent receives a reward of 1.0 when reaching the goal, and a negative reward for collisions with walls, which is dependent on the agent's speed.

**ProcTHOR**: We sample 336 training and 256 evaluating houses from the ProcTHOR-10K dataset. Unless otherwise specified, we keep both the trajectories and the environments of the validation datasets in ProcTHOR and Mazes separate from those of the training datasets. In seen-task validation, the environments have overlaps, but the trajectories are resampled.

### B.2 Details of Model Structures

The framework consists of the following modules:

- Observation Encoder (Image): A convolutional encoder that processes 128×128 images into a $D$-dimensional latent vector through 9 convolution layers and 8 residual blocks.
- Observation Decoder (Image): A deconvolutional decoder that reconstructs 128×128 images from a $D$-dimensional latent vector through 1 convolution layer, 8 residual blocks, and 3 transposed-convolution layers.
- Observation Encoder (Raw): A linear layer mapping from hidden size of $4$ to $D$.
- Observation Decoder (Raw): A linear layer mapping from hidden size of $D$ to $4$.
- Latent Decoder: A 1-layer MLP with input size $D$, hidden size $D$, layer normalization, and residual connections.
- Action Encoder: A 1-layer MLP that encodes discrete actions into a $D$-dimensional hidden state.
- Sequence Decoder: A gated self-attention architecture with $L$ layers, hidden size $D$, inner hidden size $D$, $d$ attention heads, memory length $l_M$, layer normalization, and block recurrence.

The hyper-parameters are specified as Table 4.

### B.3 Details of Training

All models were trained on NVIDIA A800 GPUs using the AdamW optimizer with the default settings in PyTorch (Paszke et al., 2019).

**Cart-pole training details.** We train the model with a per-GPU batch size of 128, an epoch of 100, an initial learning rate of $1.0\mathrm{e}^{-4}$ and decayed to $2.04\mathrm{e}^{-5}$.

**Maze pre-training details.** We first train the Image Encoder and Image Decoder with a per-GPU batch size of 400, an epoch of 50, and an initial learning rate of $3\mathrm{e}^{-4}$. Subsequently, we train all model components with a per-GPU batch size of 10, an epoch of 10, an initial learning rate of $2.0\mathrm{e}^{-4}$, and decayed to $8.8\mathrm{e}^{-5}$.

**ProcTHOR training details.** We first train the Image Encoder and Image Decoder using the same settings as in Maze pretraining, except for the initial learning rate, which is set to $2.0e^{-4}$. The VAE is then frozen, while the remaining modules are initialized from the weights trained on the Maze dataset and further fine-tuned on ProcThor data. For the 40k version of ProcThor, we train for 10 epochs with a per-GPU batch size of 10 and an initial learning rate of $2.0e^{-4}$, decayed to $1.2e^{-4}$. For the 5k version, we use the same settings but train for 20 epochs.

## C  ADDITIONAL RESULTS

**Additional reults of the performances of L2World in Mazes.** Table 5 presents the prediction error (measured as the mean square error between observations) of L2World and the other baselines on the Maze datasets. The prediction errors are fully consistent with the PSNR evaluation; therefore, we report only the PSNR results in the remaining experiments. Table 6 details the K-step prediction performance, corresponding to the histograms in Figure 4.

Table 5: 1-step prediction error($\downarrow$) of different world models in Mazes.

| Model | Seen | | | | | Unseen | | | | |
|---|---|---|---|---|---|---|---|---|---|---|
| | T=1 | T=10 | T=100 | T=1000 | T=10000 | T=1 | T=10 | T=100 | T=1000 | T=10000 |
| L2World (Maze-32K-L) | $2.09 \times 10^{-2}$ | $8.01 \times 10^{-3}$ | $4.89 \times 10^{-3}$ | $3.43 \times 10^{-3}$ | $3.13 \times 10^{-3}$ | $2.30 \times 10^{-2}$ | $\mathbf{7.51 \times 10^{-3}}$ | $\mathbf{4.82 \times 10^{-3}}$ | $\mathbf{3.42 \times 10^{-3}}$ | $\mathbf{3.42 \times 10^{-3}}$ |
| L2World (Maze-32K-S) | $1.39 \times 10^{-2}$ | $1.18 \times 10^{-2}$ | $1.08 \times 10^{-2}$ | $9.53 \times 10^{-3}$ | $8.96 \times 10^{-3}$ | $\mathbf{1.43 \times 10^{-2}}$ | $1.19 \times 10^{-2}$ | $1.09 \times 10^{-2}$ | $9.35 \times 10^{-3}$ | $9.31 \times 10^{-3}$ |
| L2World (Maze-128-S) | $1.13 \times 10^{-2}$ | $9.15 \times 10^{-3}$ | $8.74 \times 10^{-3}$ | $6.28 \times 10^{-3}$ | $6.66 \times 10^{-3}$ | $1.58 \times 10^{-2}$ | $1.37 \times 10^{-2}$ | $1.26 \times 10^{-2}$ | $1.08 \times 10^{-2}$ | $1.09 \times 10^{-2}$ |
| L2World (Maze-128-L) | $1.40 \times 10^{-2}$ | $8.20 \times 10^{-3}$ | $\mathbf{4.66 \times 10^{-3}}$ | $\mathbf{2.72 \times 10^{-3}}$ | $\mathbf{2.51 \times 10^{-3}}$ | $1.76 \times 10^{-2}$ | $1.14 \times 10^{-2}$ | $8.02 \times 10^{-3}$ | $7.01 \times 10^{-3}$ | $7.04 \times 10^{-3}$ |
| Dreamer (Maze-32K-L) | $2.29 \times 10^{-2}$ | $\mathbf{6.57 \times 10^{-3}}$ | $1.19 \times 10^{-2}$ | $7.47 \times 10^{-3}$ | $6.47 \times 10^{-3}$ | $2.08 \times 10^{-2}$ | $8.95 \times 10^{-3}$ | $7.25 \times 10^{-3}$ | $5.43 \times 10^{-3}$ | $6.14 \times 10^{-3}$ |
| Dreamer (Maze-128-L) | $1.94 \times 10^{-2}$ | $8.63 \times 10^{-3}$ | $6.56 \times 10^{-3}$ | $6.02 \times 10^{-3}$ | $5.72 \times 10^{-3}$ | $3.75 \times 10^{-2}$ | $3.51 \times 10^{-2}$ | $3.90 \times 10^{-2}$ | $4.50 \times 10^{-2}$ | $4.46 \times 10^{-2}$ |
| NWM (Maze-32K-L) | $\mathbf{8.22 \times 10^{-3}}$ | $9.52 \times 10^{-3}$ | $1.20 \times 10^{-2}$ | $5.86 \times 10^{-3}$ | $7.83 \times 10^{-3}$ | $2.02 \times 10^{-2}$ | $2.26 \times 10^{-2}$ | $2.48 \times 10^{-2}$ | $1.70 \times 10^{-2}$ | $1.28 \times 10^{-2}$ |

Table 6: Average PSNR of $k$-Step auto-regressive prediction in Mazes (Unseen).

| Model | T=1 | | | | T=10 | | | | T=100 | | | | T=1000 | | | | T=10000 | | | |
|---|---|---|---|---|---|---|---|---|---|---|---|---|---|---|---|---|---|---|---|---|
| | $k=1$ | $k=2$ | $k=4$ | $k=8$ | $k=1$ | $k=2$ | $k=4$ | $k=8$ | $k=1$ | $k=2$ | $k=4$ | $k=8$ | $k=1$ | $k=2$ | $k=4$ | $k=8$ | $k=1$ | $k=2$ | $k=4$ | $k=8$ |
| L2World (Maze-32K-L) | 16.94 | 15.79 | 14.74 | 13.39 | **21.02** | **19.88** | **18.96** | 15.83 | **23.24** | **22.16** | **20.85** | **18.31** | **24.60** | **23.57** | **21.33** | **18.03** | **24.80** | **24.11** | **21.81** | **18.72** |
| L2World (Maze-32K-S) | **18.44** | **17.41** | **16.82** | **15.61** | 19.23 | 19.05 | 17.92 | **16.73** | 19.62 | 19.43 | 18.19 | 16.69 | 20.29 | 20.17 | 18.89 | 17.03 | 20.31 | 20.21 | 18.83 | 17.00 |
| L2World (Maze-128-S) | 18.00 | 16.83 | 16.23 | 15.17 | 18.62 | 18.18 | 17.06 | 15.96 | 18.98 | 18.53 | 17.39 | 15.93 | 19.65 | 19.09 | 17.86 | 16.10 | 19.62 | 19.12 | 17.91 | 16.18 |
| L2World (Maze-128-L) | 17.53 | 15.99 | 15.38 | 14.53 | 19.45 | 18.73 | 17.67 | 16.64 | 20.96 | 20.47 | 19.11 | 17.57 | 21.54 | 20.76 | 19.45 | 17.48 | 21.53 | 21.05 | 19.58 | 17.65 |
| Dreamer (Maze-32K-L) | 16.16 | 15.80 | 14.62 | 13.12 | 19.66 | 18.96 | 17.72 | 16.25 | 21.89 | 20.99 | 20.33 | 17.74 | 23.13 | 22.08 | 20.57 | 17.00 | 23.16 | 21.85 | 20.23 | 16.14 |
| NWM (Maze-32K-L) | 16.20 | 13.37 | 11.68 | 11.16 | 16.71 | 13.34 | 11.65 | 11.00 | 17.00 | 13.86 | 12.13 | 11.48 | 17.37 | 13.82 | 12.24 | 12.60 | 17.85 | 14.51 | 12.60 | 12.46 |

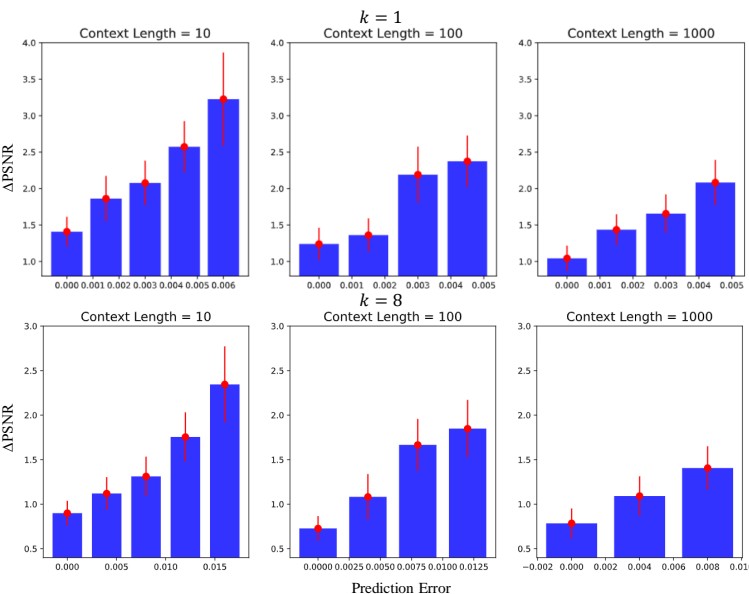

Figure 10: We investigated the correlation between the average prediction error and the performance change. The performance change was quantified by $\Delta$PSNR. $\Delta$PSNR is acquired by the loss of accuracy by replacing the input $s_t$ with predicted $\hat{s}_t$ and then measuring the loss of PSNR in future predictions with $k = 1$ and $k = 8$.

**EL and ER in navigation world models implicitly perform global mapping.** We further show that predicting transitions alone, without any specifically designed tasks, can potentially capture the global map implicitly. We collect 12 trajectories from 4 unseen mazes (3 trajectories in each maze) and track the transformation of memory states ($\phi_t$) across each of the linear attention layers. Among the 18 layers used in the transition model, we select layers $\{1, 6, 12, 18\}$ for t-SNE visualization of the memory states ($\phi_t$). To ensure that the 4 unseen mazes are not trivially discriminable, for instance, by geometric and embodiment configurations in one-shot, we maintain identical configurations for most aspects of the evaluated 4 mazes while varying only their topology, such as the arrangement of walls. We employ silhouette scores (Rousseeuw, 1987) to quantify clustering quality, where higher values indicate better environment separation. As visualized in Figure 11, we highlight two insights: first, training solely on transition prediction across diverse environments and long contextual windows implicitly learns a spatial map, potentially removing the need for auxiliary mapping modules; second, EL and ER behave differently across layers, suggesting distinct underlying mechanisms.

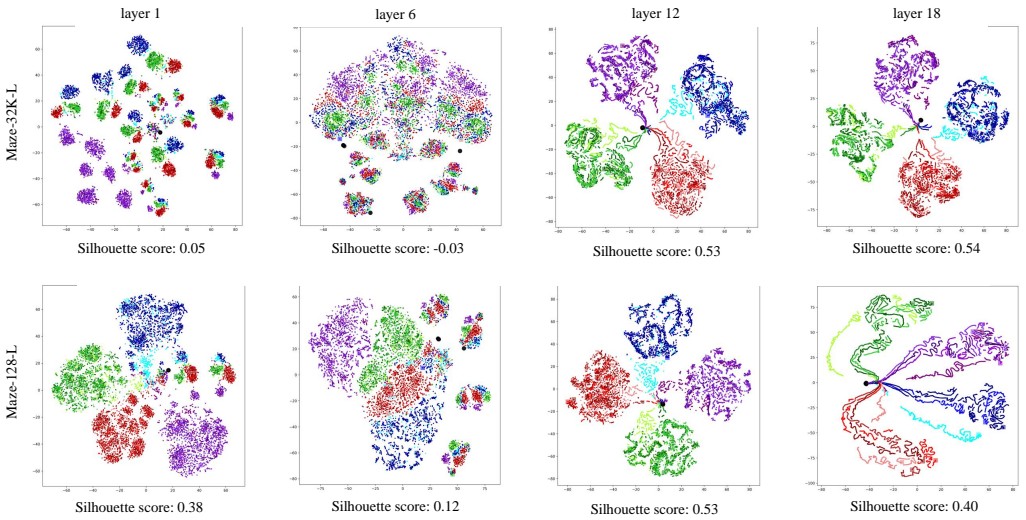

Figure 11: t-SNE visualization of the memory states $\phi_t$ in state transition prediction for Maze-32K-L and Maze-128-L groups. We visualize the memory states from layers 1, 6, 12, and 18. The visualization encompasses 12 trajectories across 4 distinct environments that are similar yet exhibit slight differences. Trajectories originating from the same environment are indicated by similar colors. The Silhouette score is computed by treating trajectories from each environment as individual classes.

**Investigating EL from perspectives of predictive coding**. The concept of predictive coding has emerged as a foundational mechanism in both biological and artificial learning systems (Rao & Ballard, 1999), in which the discrepancy between expected and observed outcomes drives attention and learning. To investigate the correlation between EL and predictive coding, we conducted experiments to examine how prediction error influences learning progress. Specifically, we selected three positions $T = \{10, 100, 1000\}$ for each of 256 evaluating sequences. At these positions, we replaced the ground-truth observations $s_t$ with model-generated predictions, thereby suppressing error-correction signals derived from real-world feedback. We then compared the accuracy loss in subsequent frames (including $k = 1$ and $k = 8$) with and without this replacement. This comparison quantifies the importance of ground-truth observations for learning progress. As illustrated in Figure 10, we observed a clear positive correlation between the mean performance difference ($\Delta$PSNR) and the prediction error (MSE loss) of the replaced frame. This finding suggests that EL progress is sensitive to prediction error, a phenomenon reminiscent of predictive coding in biological systems. These results further validate ICL as a prospective mechanism for adapting world models to variant environments.

**Cases**. Figure 12 illustrates 10-step-ahead predictions at $T = \{1, 100, 10K\}$ for our method, Dreamer, and NWM, all trained on Maze-32K-L. NWM produces visually convincing frames with

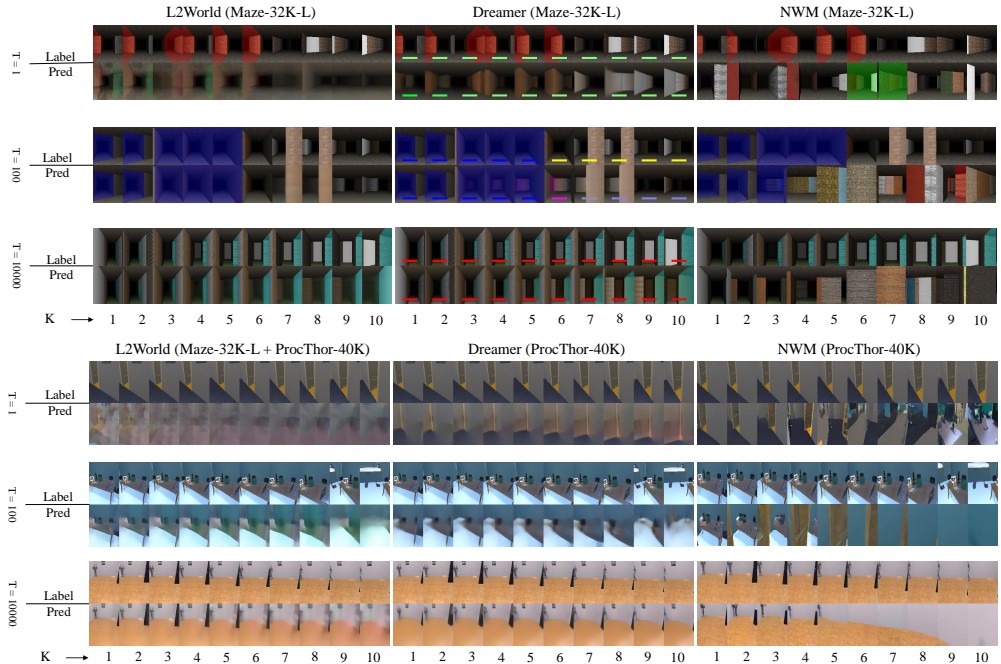

Figure 12: Example predictions produced by L2World and the baselines in Mazes and ProcTHOR for $T = \{1, 100, 10000\}$ and $k = 10$, together with the corresponding ground-truth sequences.

fine textures and a plausible layout; however, frame-wise fidelity alone is insufficient for accurate long-range forecasting because the model lacks long-term memory and spatial reasoning. Figure 13 juxtaposes the performance of L2World trained with 32K-L and 32K-S, clearly demonstrating the benefit of longer sequences in promoting EL. Figure 14 shows two failure cases of L2World (Maze-32K-L) on unseen maze environments. These failures appear to result from excessive blurring in the predictions, causing the compound error to escalate rapidly as $k$ increases. Building on this observation, a natural extension of our work is to incorporate additional overshooting during training so that the world model can better forecast distant futures.

# D   USE OF LLMS

We used large language models (LLMs) only as an auxiliary tool to improve the clarity and presentation of this paper. The assistance was limited to:

- **Language refinement:** grammar checking, wording suggestions, and improving sentence fluency while preserving the authors' original technical content.
- **Mathematical support:** helping verify the correctness and readability of some derivations and notations, without introducing new technical results.

No LLM was used for generating research ideas, designing experiments, analyzing results, or writing original scientific content. All conceptual and technical contributions were made by the authors.

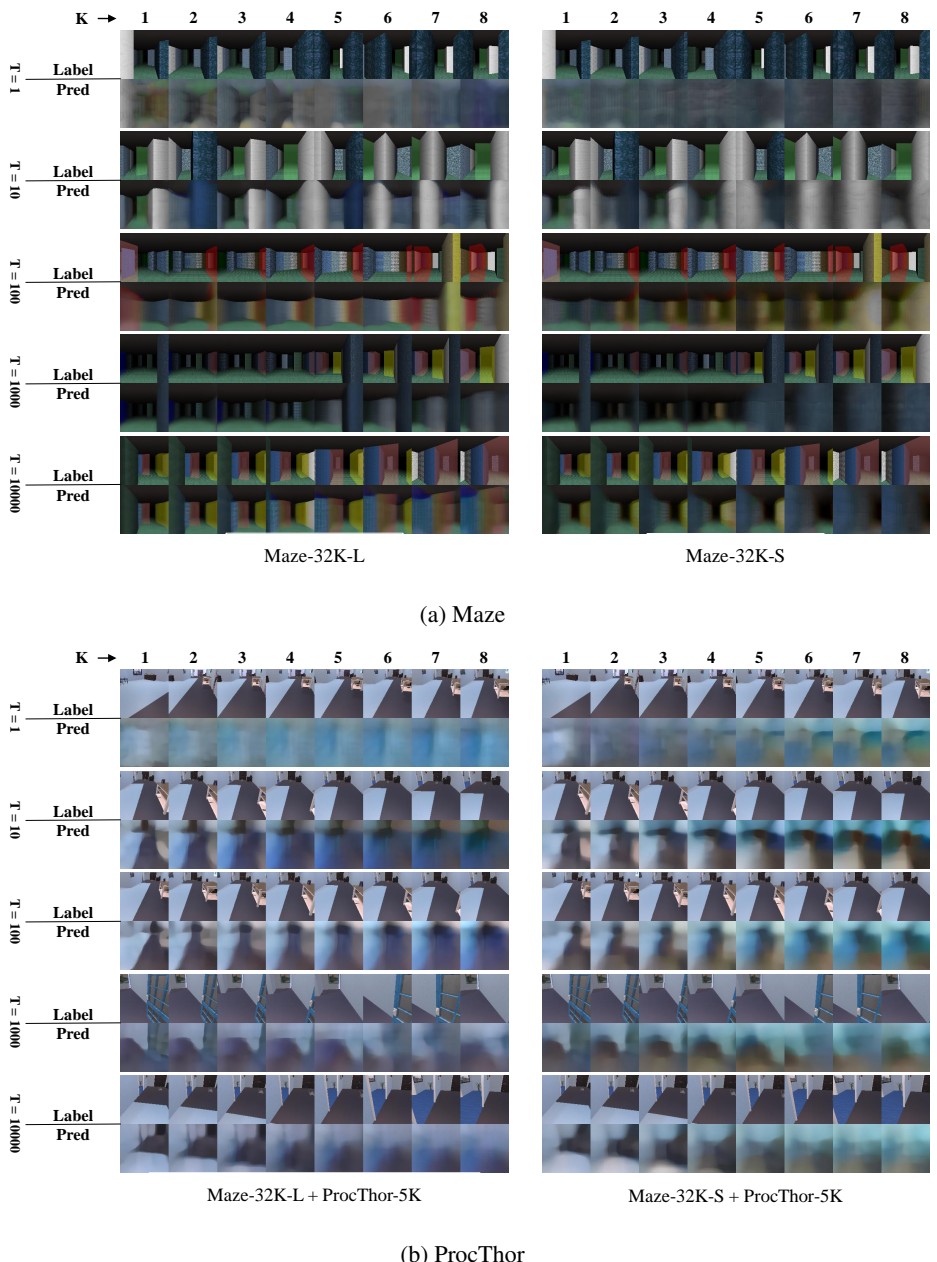

(a) Maze

(b) ProcThor

Figure 13: Example predictions by L2World trained with Maze-32K-L and Maze-32K-S at $T = \{1, 10, 100, 1000, 10000\}$ and $k = 8$.

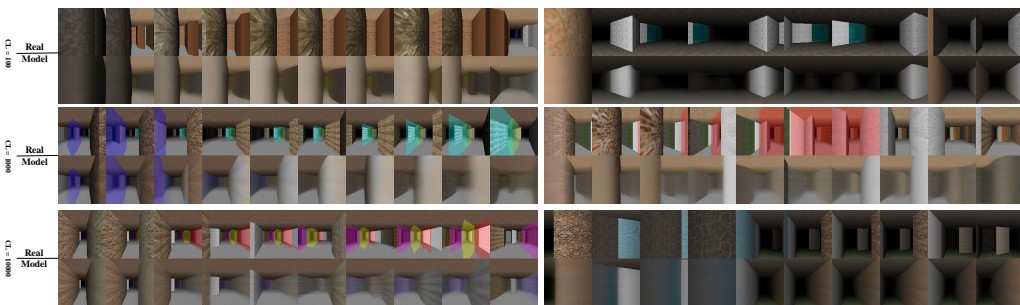

Figure 14: 2 failed examples produced by L2World on Mazes with $T = \{1, 100, 10000\}$ and $k = 8$.

