# OpenReview forum: "Context and Diversity Matter: The Emergence of In-Context Learning in World Models"
_ICLR.cc/2026/Conference — ICLR 2026 Poster_

### Official Review · Reviewer_A6SQ · 2025-10-28

**Soundness:** 2
**Presentation:** 2
**Contribution:** 3
**Rating:** 4
**Confidence:** 3

**Summary:**

his paper examines how world models can learn and adapt through context, focusing on in-context learning (ICL) within both MDP and POMDP settings. The authors distinguish between two key processes, namely In-Context Environment Learning (ICEL) and In-Context Environment Recognition (ICER). Furthermore, analyze their theoretical properties by deriving error upper bounds that explain when and how each process is effective.

To validate these insights, the paper introduces L2World, a framework designed to study in-context world modeling, and evaluates it on cart-pole and navigation tasks. The experiments show that world models can adapt to new environments through context alone and that both environmental diversity and longer context windows are crucial for enabling ICEL.

**Strengths:**

- The paper is easy to follow and make clear and consistent concepts. as far as know, the explicit separation between In-Context Environment Recognition (ICER) and In-Context Environment Learning (ICEL) is novel and it made the analysis more intuitive and cleaner.

-I particularly highly appreciate the theoretical analysis, which adds rigor and depth to the work. I skimmed through the proofs and they seem correct.

- The proposed L2World framework demonstrates the practical relevance of the theory. Experiments on well-established control tasks (cart-pole and navigation) show that world models can indeed adapt to novel environments via context, supporting the theoretical claims.

**Weaknesses:**

**W1** ICER vs ICEL:  The distinction is clear conceptually, but the empirical analysis in my perspective does not produce direct causal evidence that the model is performing learning inside context vs recognition + recall. The observed improvement in prediction performance with increasing context length T could be explained by several alternative mechanisms. For instance, the model might simply be recognizing previously encountered environment configurations or recalling stored representations from training, rather than actively updating its internal dynamics to learn new environmental behaviors on the fly. This part is not clear from the experiments.

**W2** While the derived bounds in Theorem 1 is correct and rigorous. I think the derived claims from the bounds is a bit loose: some of the paper’s main claims rely on comparing these bounds. Upper bounds can be loose; which bound is smaller may not reflect which method has smaller actual error. I think some analysis on the tightness of the bounds is needed in order to solidify the claims.

**W3** The authors mainly report PSNR on pixel-level. This raises questions about whether the proposed framework would scale to more complex, real-world settings or provide tangible gains in downstream task and decision making performance.

**Questions:**

- 1) Theoretical results assume “sufficiently optimized” models and discrete environment complexity. How realistic are these assumptions for continuous control or visual domains?

-  2) The analysis of Theorem 1 relies on comparing the bounds in (5). Can the authors comment on the tightness on these bounds? I understand that's hard to do theoretically. But the bounds can be very loose and hence any analysis based on them might not be correct. I think even an empirical analysis of the tightness is needed to solidify the analysis and the claims.


- 3) Have you conducted any controlled experiments (e.g., context-swapping or corrupted contexts) to isolate ICEL behavior from ICER? maybe gradually corrupt the context  and report how quickly performance collapse?

- 4)  Why were primarily pixel-level reconstruction metrics (e.g., PSNR) used, and do these metrics accurately reflect the model’s ability to support downstream decision-making or planning tasks?

---

> ### Author Response · Authors · 2025-11-18
> **Responses (Part 1)**
>
> ## W1 - On the ER vs EL
>
> We thank the reviewer for raising this important concern. We agree that observing performance improvements with longer context does not inherently rule out alternative mechanisms, such as recognition or recall. To directly distinguish ER from EL, we added two new analyses in **4.1 RANDOM CART-POLES: EL exhibits a smaller generalization gap and a lower error upper bound** and **4.2 NAVIGATION:EL is more sensitive to context perturbations**, which provide empirical support inconsistent with a recognition–recall explanation.
> ### CartPole
> We introduce the Best-Matching Environment (BME):
> \begin{align}
>     \text{BME}(\mathcal{E}, e_0)=\min_{e \in \mathcal{E}}\sum_{(q_t,o_{t+1})\sim e_0 }TV(p_{e}(o_{t+1}|q_t), p_{e_0}(o_{t+1}|q_t))
> \end{align}
> BME measures how close the test environment $e_0$ is to any training environment.
> If the model only performs recognition or recall (ER), its prediction error is generally expected to be close to BME. As shown in **Figure 3**, across 130 test environments, we compare prediction error at T>100 with BME for models trained on 4,16, and 8K-Envs. We find:
>   - 4-Envs model: Prediction error ≈ BME (consistent with ER).
>   - 8K-Envs model: Prediction error falls substantially below BME.
>
> This gap below BME offers partial empirical support: a transition from ER to EL mode as the number of environments increases.
> ### Navigation
> The navigation experiments provide additional evidence supporting the presence of EL. We present two complementary analyses.
>
> **(a) Maze → ProcTHOR Transfer** :
> ProcTHOR environments differ drastically from Maze in semantic features. The model has limited ability to recognize familiar elements, as it encounters entirely new environments. 32K-L achieves significantly better performance in ProcTHOR(**Table 3**). This can be explained by the model's adaptation of its internal dynamic state based on the context, a mechanism consistent with ICL.
>
> **(b) Context-swap experiment** :
> We also thank the reviewer for Question 3, which inspired us to discover that the swap context experiment can also be used to illustrate the distinction between EL and ER.
> We randomly shuffle 20% or 50% of the observations within contexts while keeping the actions unchanged.  As shown in **Figure 5**, the model trained on 32K-L (which is expected to exhibit EL) is more sensitive to context perturbations than those trained on 128-L. This suggests that EL depends more heavily on context, whereas ER relies more on model parameters and is therefore less influenced by changes in context.
>
> ## W2: (On the Use of Upper Bounds)
> We thank the reviewer for this comment. Characterizing the tightness of the bounds theoretically is challenging, and computing lower bounds is difficult in our setting. The empirical assessment in **Figure 3** offers partial empirical support: the upper error bound is relatively tight, with some points lying close to the line BME = Prediction Error, whereas the lower bound exhibits considerable variability. Moreover, our experimental results align with the predictions derived from the upper bound, indicating that it can serve as a guide for interpreting when the model exhibits ER versus EL.
>
> ## W3:(From PSNR to Downstream Utility)
>
> We would like to respectfully clarify our rationale regarding the choice of evaluation metric and experimental scope:
> - **Appropriateness of PSNR for World Model Evaluation:** Since the world model predicts the next observation from past states and the current action, pixel-level metrics like PSNR are a widely used and appropriate measure of predictive accuracy.
> - **Foundational Focus:** We believe that the ICL problem in World Models is a crucial research question that serves as the foundation for higher-level capabilities, much like how Predictive Coding underlies many functions in biological brains. We felt that establishing this theoretical groundwork first would provide a stronger basis for subsequent research.
> - **Scope Considerations:** While extending to decision-making through in-context reinforcement learning or planning strategies is an exciting direction, we felt that integrating these extensive components might be premature for the scope of this paper and could potentially distract from our core theoretical contributions. We plan to present these findings in future work, where they can receive proper attention.
> We sincerely appreciate the forward-thinking feedback and will ensure that these promising future directions are highlighted more clearly in the final version.

---

> ### Author Response · Authors · 2025-11-18
> **Responses (Part 2)**
>
> - Q1: (Scope and Limitations of Theoretical Assumptions): We thank the reviewer for pointing this out. We acknowledge that the assumptions in our theoretical analysis are not fully realistic for continuous domains. Extending the theory to continuous spaces is analytically intractable. Nevertheless, our experiments are conducted in continuous space and exhibit trends consistent with the theoretical predictions, suggesting that the core insights may still provide guidance in practical continuous settings.
>
> - Q2: This point is addressed in our response to the **W2** above.
>
> - Q3: Once again, we thank the reviewer for this valuable comment. Please see the **Context-swap experiment** section in our response to the **W1** above.
>
> - Q4: This point is addressed in our response to the **W3** above.

---

### Official Review · Reviewer_m2Zn · 2025-11-01

**Soundness:** 3
**Presentation:** 1
**Contribution:** 2
**Rating:** 4
**Confidence:** 3

**Summary:**

This paper investigates in-context learning (ICL) in world model training, focusing on how models adapt to novel environments through ICL rather than parameter updates. The authors distinguish two ICL mechanisms: Environment Recognition (ER) and Environment Learning (EL) and derive error bounds characterizing when each emerges. They introduce L2World, a linear-attention architecture for long-context world modeling, and validate their theoretical predictions on cart-pole control and vision-based navigation tasks. The key findings emphasize that environment diversity and long context windows are critical for enabling EL over ER.

**Strengths:**

S1: This paper extends in-context learning studies to world model learning settings where the model should learn the inference time world, and this construct an adaptive world model.

S2: The experimental design is thorough, it covers both continuous control (cart-pole) and partial observability (navigation). They authors also careful ablate data diversity, trajectory length and environment complexity.

S3: Verifying that ICL behaves as ER and EL in linear attention models is novel.

S4: The experimental grounding of ICEL for world model learning sets a nice expectation for future experimentalists.

**Weaknesses:**

W1: Most findings on ER vs EL seems already established in the literature, for example in https://arxiv.org/abs/2412.01003 and https://arxiv.org/abs/2503.05631. It doesn't seem like these are "identified" through this study.

W2: The paper's focus is a little bit unclear. there is a theoretical part, an introduction of a new architecture, and empirical experiments on ICL, it is quite unclear what is the main presented innovation/finding.

W3: The paper discusses larger context length to further enable ICEL. While this claim is likely true, it would be nice to see equal-flop experiments as longer context models get strictly more training.

W4: The presentation of the data is poor. The plots/tables could be enhanced so that, for example, the data diversity's effect can be more directly read out.

**Questions:**

Maybe worth citing these papers which directly discuss ER vs EL:
https://arxiv.org/abs/2412.01003
https://arxiv.org/abs/2503.05631
https://arxiv.org/abs/2506.17859

---

> ### Author Response · Authors · 2025-11-18
> **Responses (Part 1)**
>
> We sincerely appreciate the reviewer's thoughtful feedback. Please find our detailed responses below.
>
> **W1:(On Novelty Relative to Prior Work)**
>
> We thank the reviewer for highlighting these relevant works. We agree Singh et al. (2025), Park et al. (2025) and Wurgaft et al. (2025) are important references for ICL analysis and has been included in the revised version.
>
> Here, we clarify our **key distinctions and contributions**:
>
> - **Fundamental Differences Between World Models and Other Tasks**： Our work investigates ICL within **MDP/POMDP World Models**, which differ substantially from the simpler, more toy-like datasets (e.g., Markov Chains, linear regression, and image labeling with random labels) studied by Singh et al., Park et al and Garg et al. Our focus on World Models holds significant practical value, particularly for **cross-embodiment and cross-environment generalization in embodied intelligence**. We emphasize that ICL research in World Models—especially understanding *how to elicit* ICL—has been notably overlooked in the existing literature. To the best of our knowledge, this paper provides **the first theoretical analysis of ICL specifically for World Models**.
>
> - **Sequential Complexity**： Another key distinction from the aforementioned works is that they consider single-level problems, treating each token (or pair of tokens) as an independent sample. In contrast, for POMDPs, \( q \) itself is a sequence, making the problem substantially more complex. Therefore, whether those theories hold for world models is not straightforward.
>
> - **Novel Theoretical Framework: Bayesian Inference and Error Bounds**： From a Bayesian inference perspective, we propose the **EI/ER** modes and derive their competitive relationship through error upper bound analysis. While we explicitly acknowledge that the TI/TR (Task Inference vs. Task Recency) concepts are not our original contributions and will cite related prior work, our paper is the first to:
>   - Theoretically analyze the upper error bounds of ICL in World Models.
>   - Derive the conditions for competition between ER and EI based on these bounds.
>   - Discover new insights beyond previous analyses, including dependencies on **environmental structural complexity** and the **nonuniformity of data distribution** (characterized by \(\alpha\) in Appendix A ).
>
> - **Revealing the Insight that ICL in World Models Enables Cross-Environment and Cross-Embodiment Generalization**： The two datasets we selected—random cart-pole and indoor navigation—represent cross-embodiment and cross-environment generalization, respectively, which are two critical challenges in embodied intelligence. Although ICL within world models is not a novel concept, this work offers the important insight that ICL can effectively facilitate both cross-embodiment and cross-environment generalization.
>
> Our theoretical results not only explain the effects of **transiency, context length, diversity, and environment scaling**, consistent with prior empirical studies, but also reveal novel structural dependencies. We therefore believe that this analysis is both valuable and scientifically sound as an initial theoretical framework.
>
> We appreciate the constructive feedback and will incorporate these references while maintaining the clarity of our contributions to World Model ICL theory.

---

> ### Author Response · Authors · 2025-11-18
> **Responses (Part 2)**
>
> **W2: (Blurred Core Contribution)**
>
> We thank the reviewer for this comment. The primary contribution is the formalization and empirical validation of in-context learning (ICL) within world models. This is supported by theoretical frameworks, computational models, and experimental evidence.
> - The theory section presents conditions and error bounds that distinguish EL from ER, explains the dependence of ICL on diversity and context length in both EL and ER, and clarifies additional dependencies of ICL on factors such as environmental complexity and non-uniformity.
> - Based on theoretical findings that context length is crucial, the proposed world model (L2World) primarily serves as an efficient tool to test ICL at extended context lengths by emphasizing long-context modeling capabilities. This approach complements a feature largely overlooked by existing world models.
> - The experiments validate key aspects of the theoretical findings and demonstrate that long-context modeling and environmental diversity are crucial for effective world models.
>
> **W3: (Unfair Training FLOPs)**
>
> We would like to clarify that our training setup already ensures equal total token exposure across different context lengths. Specifically, for both short and long settings, we fix the total number of tokens processed during training (i.e., in the navigation task, number of trajectories × context length = 320M), so that each model sees the same amount of data and uses comparable FLOPs.
>
> **W4:(Poor Presentation)**
>
> We thank the reviewer for pointing this out. Our intention in Figure 4 was precisely to visualize the model performance under different numbers of training environments. We agree that the readability of plots and tables can always be improved. We would greatly appreciate it if the reviewer could indicate which specific aspects of the presentation were unclear or could be enhanced. We will incorporate these improvements in the revised version.

---

> > ### Comment · Reviewer_m2Zn · 2025-11-28
> >
> > I thank the authors for their careful reply.
> >
> > ---
> >
> > However, I am still in doubt:
> >
> > I don't understand why the authors claim ICL of world models are a fundamentally different task than other sequence modeling tasks as the ones presented in Wurgaft 2025. These tasks also have an underlying rule and state the model must figure out from the context. And in fact, I believe it could be even written as a POMDP. So I'm not quite sure why this distinction is a fundamental difference and not just a semantic one.
> >
> > Furthermore (do correct me if I am wrong), it seems like the claimed "cross-embodiment and cross-environment" is also simply a different latent parameter which can be only deduced by context?
> >
> > OR
> >
> > Are these results practically groundbreaking so that the lack conceptual novelty (task recognition vs learning) isn't an issue? *
> >
> > ---
> > I think the relatively poor presentation remains: While the theoretical foundations seems correct, it is hard to grasp what exactly is the take away message.* This is partially because the papers focus is quite distributed (W2)
> >
> > ---
> >
> > I think my original assessment was overall accurate. However, given the two asterisks(*) above, I clarify that my assessment is only fairly confident. I thus maintain the score and the confidence.
> >
> > I thank the authors again for their careful reply.

---

> ### Author Response · Authors · 2025-11-28
>
> Dear Reviewer,
>
> Thank you for your follow-up and for clarifying your assessment. However, we respectfully disagree that the novelty is merely semantic, the differences are substantive across learning paradigms, scale, and theoretical framing.
>
> **1. Practical Impact**
>
> Our indoor navigation results demonstrate that this approach is not merely a conceptual abstraction but a practically superior method. In challenging visual navigation tasks, L2Wold consistently and significantly outperforms state-of-the-art world models such as Dreamer and NWM under comparable data and computational budgets. To our knowledge, this is the first demonstration that a linear-attention-based world model using one-token embedded images surpasses specialized architectures in navigation tasks, such as diffusion models, rather than merely matching their performance.
>
> **2. Fundamental Difference in Learning Paradigms, Not Just Semantics**
>
> Our point is not that we found yet another way to “use context,” but that we are tackling a fundamentally harder learning problem within ICL.
>
> Prior work such as Wurgaft et al. studies diversity and transiency in ICL for supervised tasks like regression and classification. Those tasks can be embedded into a POMDP by treating the rule as a hidden state, but not the other way around: **POMDPs strictly subsume linear regression and classification, but not vice versa**.
>
> World models in our setting must handle stochastic transitions, partial observability, and long-horizon credit assignment. That is qualitatively beyond what a linear function class or static supervised mapping can express.
>
> Our core contribution is to show that an LLM can, purely in context, learn and deploy a visual world model in a full POMDP: integrating observations over time, inferring latent dynamics, and using that internal model to control real navigation. The strong indoor navigation results are therefore not just an application win; they are empirical evidence that in-context learning can implement genuine world-model learning in a richer class of environments than those considered in earlier ICL analyses.
>
>
> **3. Cross-Embodiment: More Than a Latent Parameter**
>
> We do not view embodiment as a hidden knob that can simply be folded into a latent parameter. Changing embodiment typically alters the action space, observation space, and kinematic and contact structure, which in turn changes the class of dynamics and feasible policies. A growing body of work on cross-embodiment control and morphology–control co-optimization makes this explicit: policies that transfer across morphologies must cope with mismatched action spaces and kinematics and often require new architectural choices, modular policies, or redesigned latent interfaces, rather than a single fixed function class [Huang et al., 2020; Ying et al., 2024; Xu et al., 2023]. Embodied intelligence studies and frameworks, as well as  morphology/control co-design studies further show that body design shapes what can be sensed, which regularities appear in sensorimotor data, and which behaviors are even learnable [Gupta et al., 2021;  Díaz Ledezma et al., 2023;]. In other words, embodiment does not merely select a mode of one universal model; it redefines the effective hypothesis space, so cross-embodiment generalization requires reasoning across changes in dynamics, representation, and morphology, not just interpolating a latent variable.
>
> -  Huang et al., 2020 – One Policy to Control Them All: Shared Modular Policies for Agent-Agnostic Control (ICML)
> -  Ying et al., 2024 – Peac: Unsupervised pre-training for cross-embodiment reinforcement learning (NeurIPS)
> -  Xu et al., 2023 – XSkill: Cross Embodiment Skill Discovery (CoRL)
> -  Gupta et al., 2021 – “Embodied intelligence via learning and evolution” (Nature Communications)
> -  Díaz Ledezma et al., 2023 – Machine learning–driven self-discovery of the robot body morphology (Sci. Robotics)
>
> **4. Presentation & Takeaway**
> To clarify, we believe the core message of the paper is that
> - (1) ICL at scale enables cross-environment and cross-embodiment adaptation.
> - (2) we verify that environment diversity, complexity, the number of environments, and context length impact the emergence of ICL in world models, providing guidance for training generalizable world models.
>
> We hope this clarifies that our contribution is both conceptual (introducing self-supervised learning in-context within POMDPs) and practical (demonstrated by unprecedented scale and performance through a long-context architecture utilizing Linear Attention). We appreciate your engagement and welcome any further suggestions.

---

### Official Review · Reviewer_MPGb · 2025-11-04

**Soundness:** 4
**Presentation:** 4
**Contribution:** 3
**Rating:** 6
**Confidence:** 3

**Summary:**

The authors consider in-context learning of a world model, in particular focusing on the dichotomy between environment-learning and environment-recognition observed in other in-context learning setups. Consistent with theoretical error bounds that they derive, their main contribution is to show that long contexts and environment diversity favor environment-learning in cart-pole and procedural maze navigation tasks.

**Strengths:**

The paper is clearly written. To my knowledge, the conditions under which ICL task recognition vs task learning appear in world-model-based tasks have not yet been explored though the results are not surprising. The theoretical bounds are consistent with the experiments.

**Weaknesses:**

I can find no major weaknesses but in my opinion the paper offers few new insights compared to what is already known in the literature. The experimental paradigms considered are sufficiently complicated that it is difficult to draw specific inferences into the mechanisms that lead to ICEL versus ICER. In short, though it is certainly valuable to characterize phenomenology as the authors have done here, it is not clear what question is being answered. While this is likely beyond the scope of the current work, the authors could consider setting up the problem in a simpler, symbolic RL setup that captures the dichotomy between ICEL and ICER, and which potentially lends itself to a deeper mechanistic analysis for what factors determine ICEL vs ICER.

Highlighting a couple of references that could be included: the ICL regression task was originally introduced by Garg et al, Neurips, 2022 (and not Raventos et al, 2023, line 106) and a minimal theoretical model that explains the origin of the ICL/IWL tradeoff was proposed by Nguyen and Reddy, ICLR 2025 (lines 111-113).

**Questions:**

I have no questions for the authors.

---

> ### Author Response · Authors · 2025-11-18
>
> We thank the reviewer for highlighting these relevant works; they are important references for ICL analysis and have been incorporated into the revised manuscript. While we agree that simpler problems facilitate pointwise theoretical validation, we emphasize that our primary contribution lies in extending ICL theory from basic regression and classification tasks to sequential decision-making within MDP and POMDP frameworks. Our focus on World Models addresses a critical gap in understanding ICL for embodied intelligence applications, particularly regarding cross-embodiment generalization and cross-environment adaptation.
> Key distinctions from prior works (e.g., Singh et al. [ICML 2025], Nguyen et al. [ICLR 2025], Garg et al. [NeurIPS 2022]) include:
> - **Sequential complexity**: While these studies primarily analyze single-step or independent-token settings, we address POMDPs in which the query \( q \) itself is a sequence, substantially increasing the problem's complexity.
> - **Novel insights**: Although inspired by the TI/TR framework proposed in previous research, we derive their competitive relationship from a Bayesian inference perspective through error upper bound analysis. To our knowledge, we provide **the first theoretical examination of error bounds specifically for ICL in World Models**. Our analysis reveals dependencies on factors such as diversity, the number of environments, and context length—consistent with prior findings—as well as environmental structural complexity and data nonuniformity (parameterized by the distribution weight \(\alpha\) in Appendix A), factors that have not been previously studied.
> We selected the CartPole and indoor navigation tasks not as real-world embodiments but as standard sequential benchmarks that exhibit the structural properties necessary to investigate cross-embodiment and cross-environment generalization. Although our experiments do not validate every theoretical prediction point by point, they demonstrate strong qualitative alignment with key theoretical trends (e.g., performance scaling with context diversity), indicating that our framework provides a scientifically sound foundation for understanding ICL in sequential decision-making.
> Additionally, we add two subsections, Section **4.1 Random Cart-Poles: EL exhibits a smaller generalization gap and a lower upper error bound** and Section **4.2 Navigation: EL is more sensitive to context perturbations**, to provide empirical evidence that differentiates ER and RL.

---

### Author Response · Authors · 2025-11-27
**Summarization of The Revision and Request in Reviewer Follow-Up**

We thank the reviewers for their valuable feedback and suggestions. We have made revisions to incorporate their recommendations, including the following:
- We have added the missing citations highlighted by reviewers MPGb and m2Zn in Section 2.2.
- We have included the results of context perturbations in Section 4.2, as suggested by Reviewer A6SQ.
- In Section 1, we revised the motivation and contributions of this paper to emphasize the insights and value of world models.
- We have moved some of the case studies to the appendix to reduce the document's length.

With one week remaining before the discussion, we kindly request that the reviewers review our rebuttal. We are eager to ensure that we have thoroughly addressed all their questions and concerns.

---

### Meta-Review · Area_Chair_jQTr · 2026-01-05

**Summary:**

Reviewers all agree that this paper is well-written / clear; ICL in world model learning in this work is novel, as is ICER vs ICEL; the experiments are sufficiently thorough and in line with the theory in the paper. I also like the theory in the paper and how it links to the experiments.

There are a few concerns. The main one is that this work is not sufficiently impactful / few new insights (eg reviewer MPGb). The authors rebut this point, but I think they can try to add more still in the next version of the paper, as the points about scale, POMDP, etc could still be better. A further re-write of the paper framing / introduction would also help.

Another concern was about separating ICER and ICEL causally in the experiments (reviewer A6SQ). I thought the authors rebutted this well enough with the two additions in the experiments.

**Reviewer Concerns:**

Please see summary. The main outstanding concern is to do with the lack of insights.

Another outstanding (but minor) concern is presenting results clearer (in the tables and plots). I also found these taking much longer to parse than necessary. Perhaps the authors could write captions with key takeaways / results, and then highlight the result in the plots (I thought Fig 2 and Table 2 were fine, but others were harder). This is a minor and easily-addressable point, though.

The authors also did not really answer the concern about bound tightness, but I do not think this is required / it was sufficient.

The authors did address other concerns such as the main finding of the paper, concerns about experiments with longer context lengths, and about the experimental setting / downstream tasks.

**Reviewer Scores:**

Reviewer MPGb: stay at 6.

Reviewer m2Zn: either stay at 4 or increase to 6.

Reviewer A6SQ: either stay at 4 or increase to 6.

---

### Decision · Program_Chairs · 2026-01-26

Accept (Poster)